



# Intercomparison of biomass burning aerosol optical properties from in-situ and remote-sensing instruments in ORACLES-2016

Kristina Pistone[1,2], Jens Redemann[3], Sarah Doherty[4], Paquita Zuidema[5], Sharon Burton[6], Brian Cairns[7], Sabrina Cochrane[8], Richard Ferrare[6], Connor Flynn[9], Steffen Freitag[10], Steve Howell[10], Meloë Kacenelenbogen[2], Samuel LeBlanc[1,2], Xu Liu[6], K. Sebastian Schmidt[8], Arthur J. Sedlacek III[11], Michal Segal-Rosenhaimer[1,2], Yohei Shinozuka[2,12], Snorre Stamnes[6], Bastiaan van Diedenhoven[7], Gerard Van Harten[13], and Feng Xu[13]

[1]Bay Area Environmental Research Institute, Moffett Field, CA, USA
[2]NASA Ames Research Center, Moffett Field, CA, USA
[3]University of Oklahoma, Norman, OK, USA
[4]JISAO, University of Washington, Seattle, WA, USA
[5]University of Miami/Rosenstiel School of Marine and Atmospheric Science, Miami, FL, USA
[6]NASA Langley Research Center, Hampton, VA, USA
[7]NASA Goddard Institute for Space Studies, New York, NY, USA
[8]University of Colorado, Boulder, CO, USA
[9]Pacific Northwest National Laboratory, Richland, WA, USA
[10]University of Hawaii at Manoa, Honolulu, HI, USA
[11]Brookhaven National Laboratory, Brookhaven, NY, USA
[12]Universities Space Research Association, Mountain View, CA, USA
[13]Jet Propulsion Laboratory, Pasadena, CA, USA

**Correspondence:** Kristina Pistone (kristina.pistone@nasa.gov)

**Abstract.** The total effect of aerosols, both directly and on cloud properties, remains the biggest source of uncertainty in anthropogenic radiative forcing on the climate. Correct characterization of intensive aerosol optical properties, particularly in conditions where absorbing aerosol is present, is a crucial factor in quantifying these effects. The Southeast Atlantic Ocean (SEA), with seasonal biomass burning smoke plumes overlying and mixing with a persistent stratocumulus cloud deck, offers

an excellent natural laboratory to make the observations necessary to understand the complexities of aerosol-cloud-radiation interactions. The first field deployment of the NASA ORACLES (ObseRvations of Aerosols above CLouds and their intEractionS) campaign was conducted in September of 2016 out of Walvis Bay, Namibia.

Data collected during ORACLES-2016 are used to derive aerosol properties from an unprecedented number of simultaneous measurement techniques over this region. Here we present results from six of the eight independent instruments or instrument

combinations, all applied to measure or retrieve aerosol absorption and single scattering albedo. Most but not all of the biomass-burning aerosol was located in the free troposphere, in relative humidities typically ranging up to 60%. We present the single scattering albedo (SSA), absorbing and total aerosol optical depth (AOD and AAOD), and absorption, scattering, and extinction Ångström exponents (AAE, SAE, EAE) for specific case studies looking at near-coincident and -colocated measurements from multiple instruments, and SSAs for the broader campaign average over the monthlong deployment. For the case studies, we

find that SSA agrees within the measurement uncertainties between multiple instruments, though, over all cases, there is no



strong correlation between values reported by one instrument and another. We also find that agreement between the instruments is more robust at higher aerosol loading ($AOD_{400} > 0.4$).

The campaign-wide average and range shows differences in the values measured by each instrument. We find the ORACLES-2016 campaign-average SSA at 500nm ($SSA_{500}$) to be between 0.85 and 0.88, depending on the instrument considered
(4STAR, AirMSPI, or in situ measurements), with the inter-quartile ranges for all instruments between 0.83 and 0.89. This is consistent with previous September values reported over the region (between 0.84 and 0.90 for SSA at 550nm). The results suggest that the differences observed in the campaign-average values may be dominated by instrument-specific spatial sampling differences and the natural physical variability in aerosol conditions over the SEA, rather than fundamental methodological differences.

**1 Introduction**

Atmospheric aerosols are an important component of the climate system in terms of their direct, semidirect, and indirect radiative effects. A primary factor governing the overall magnitude of these effects is the composition, size, and concentration (and, consequently, the radiative properties) of the aerosol in a given location. While on the global average aerosols predominantly cool the planet by reflecting sunlight back to space, shortwave-absorbing aerosols (such as those from biomass burning
sources) are also capable of warming the planet by directly absorbing sunlight (called the direct effect: Chylek and Coakley, 1974; Meyer et al., 2015; Zhang et al., 2016). Aerosols can additionally affect cloud properties by microphysical indirect effects affecting droplet size and lifetime (e.g., Twomey, 1974; Albrecht, 1989; McComiskey and Feingold, 2012; Lu et al., 2018); or by altering surface evaporation, cloud burn-off rates, and atmospheric dynamics, the so-called semi-direct aerosol effects (e.g., Ackerman et al., 2000; Koch and Del Genio, 2010; Wilcox, 2010; Sakaeda et al., 2011; Wilcox, 2012; Gordon et al., 2018); or
by some combination the above (e.g., Adebiyi and Zuidema, 2018).

In this work we focus on the aerosol single-scattering albedo (SSA), the ratio of aerosol scattering to total extinction, a key intensive property which relates to the aerosol composition while being independent of the total aerosol loading. SSA is defined as the ratio of Scattering/Extinction; highly scattering particles such as sea salt will have SSA close to 1, whereas biomass burning (BB) smoke with a large fraction of soot particles will have SSA less than 1, typically between 0.7 and 0.95
(Dubovik et al., 2002), and column-average SSA, as discussed here, may be composed of contributions from several different aerosol types. SSA, among other parameters, is essential for the determination of direct aerosol radiative effects. SSA has been shown to evolve with BB plume location, age, mixing state, emission source, and distance from source (e.g., Haywood et al., 2003; Eck et al., 2013; Konovalov et al., 2017).

Other important parameters are the aerosol optical depth (AOD) and the absorbing component of AOD (AAOD, defined as
AOD*(1-SSA)), which indicate the magnitude of the column (absorbing) aerosol loading. In the present work, AODs from the different retrievals are presented to assess the similarity between temporally- and spatially-dislocated scenes observed by the different instruments, and since the retrievals of AOD and SSA are not necessarily uncoupled, AOD can be a useful diagnostic. However, as we focus here on intensive aerosol properties, the climatological AOD values are not discussed. Finally,



we discuss the Ångström exponents from each retrieval method (AAE, SAE, and EAE for the absorption, scattering, and extinction Ångström exponents, respectively). Ångström exponents are given by the log-space slope of absorption, scattering, or extinction aerosol optical depths versus wavelength, and are frequently used to characterize atmospheric aerosol. As AAE is primarily (though not entirely) determined by aerosol composition (Russell et al., 2010; Bahadur et al., 2012) and SAE is pri-

marily associated with aerosol size, these parameters can be instructive in understanding the nature of the aerosol in question. EAE is shown as well, to place these results in the context of other remote sensing results which measure extinction, though EAE closely follows SAE since extinction is dominated by scattering for almost all atmospheric aerosols. Accurate representation of the magnitude and variability of these aerosol properties on a regional scale has significant implications for aerosol radiative effects calculated using climate models and/or satellite data. As different instruments (such as those incorporated into

this work) may rely on different physical measurement principles, each with different considerations and limitations (Table 1), an understanding of how distinct observations compare to one another is a critical piece in gaining an understanding of our observational limitations for key parameters and hence in calculating aerosol effects and their uncertainties.

This paper presents data from the NASA ORACLES (ObseRvations of Aerosols above CLouds and their intEractionS) campaign (Zuidema et al., 2016), a multi-year, multi-platform collaboration to sample clouds and BB aerosol over the Southeast

Atlantic Ocean (SEA). During the 2016 ORACLES deployment, two NASA aircraft (a P-3 and an ER-2) were flown with a suite of aerosol, cloud, radiation, and meteorological instruments for remote-sensing and in-situ observations. The remainder of this section gives a brief context of previous observations over this region (Section 1.1), their implications (Section 1.2) and an overview of the ORACLES campaign (Section 1.3). Section 2 describes data and methods used, including the ORACLES instrumentation, flight paths, and comparison case criteria. In Section 3.1 we compare the two in-situ absorption measurement

techniques; Section 3.2 details two case studies of the multi-instrument comparisons (in situ and remote sensing instruments), and Section 3.3 presents campaign-wide comparisons of in-situ versus remote-sensing measurements. Finally, in Section 4 we describe the broader picture of the average aerosol properties measured over the region during ORACLES-2016 and discuss how these results fit within the context of previous observations.

## 1.1   SEA aerosol climatology

The SEA is a particularly important region in the context of understanding aerosol-radiation and aerosol-cloud interactions. This region exhibits a persistent stratocumulus cloud deck off the western coast of Africa. During the southern African biomass burning season (August to October), these clouds are situated under and/or within plumes of absorbing aerosols originating from widespread continental fires. The smoke is lofted in continental convection and then advected westward in the southerly branch of the African Easterly jet at a typical altitude of 3-5km in September (Adebiyi and Zuidema, 2016). The composition

of these aerosols (as reflected in the SSA parameter) can change the magnitude and even the sign of the radiative forcing effects of aerosol over clouds (e.g., Chand et al., 2009; Zuidema et al., 2016; Cochrane et al., in prep; Kacenelenbogen et al., 2018).

Compared with other regions of the world, there have been relatively few studies measuring aerosol properties (microphysical or radiative) either directly over the southeast Atlantic or near their source in sub-Saharan Africa. Nonetheless, there is still a good deal of previous work which can help to place the ORACLES observations in context. A key observational dataset



is from the Southern AFricAn Regional science Initiative (SAFARI 2000) campaign which used aircraft to measure aerosol properties over and in close proximity to the coast of southern Africa in September 2000, including both aged aerosol and fresh biomass plumes (e.g., Haywood et al., 2003; Schmid et al., 2003; Leahy et al., 2007; Russell et al., 2010). SAFARI 2000 also resulted in the establishment of several sites of the AErosol RObotic NETwork (AERONET) on the African continent; these

allow for longer-term climatological analysis near emission sources (e.g., Queface et al., 2003; Magi and Hobbs, 2003; Swap et al., 2003; Eck et al., 2003, 2013). Passive and active satellite observations have also been used to detect and quantify aerosol above clouds over the SEA region (e.g., Chand et al., 2008, 2009; Waquet et al., 2013; Jethva et al., 2014; Torres et al., 2012; Liu et al., 2015), though such studies typically need to assume (through lookup tables or models) cloud and aerosol properties as well as their relative geometry. In this context, the ORACLES aircraft-based dataset, designed to sample the region of

highest cloud cover and BB smoke concentration, provides important and heretofore unique observations of both aerosol and cloud over the southeast Atlantic Ocean, due to both improved instrumentation since SAFARI-2000, and because ORACLES focused on regions farther off the southern African coast than previously measured.

The previous studies give a somewhat limited yet still useful view of the temporal and seasonal trends in SSA for a limited set of locations within this region (Figure 1). In the SAFARI 2000 campaign, aircraft instrumentation was used to sample both

the aged aerosol plume (a few days old) as well as fresh biomass aerosol (a few minutes old) over Namibia and the coastal SEA. The mean SSA of the aged haze was reported by Haywood et al. (2003) as 0.91, 0.90, and 0.87 at 450, 550, and 700 nm using a combination of an in situ aircraft-based Particle Soot Absorption Photometer (PSAP) and nephelometer. However, Leahy et al. (2007) reported a lower "best estimate" (campaign-average) $SSA_{550nm}$ of $0.85 \pm 0.02$ using the SAFARI airborne flux radiometry and in situ measurements combined with ground-based AERONET retrievals (the individual flux radiometry

estimate is described in Bergstrom et al. (2003) and Russell et al. (2010) and included separately in Figure 1). For a SAFARI flight specifically targeting fresh biomass burning smoke, the reported SSA was lower, at 0.86, 0.84, and 0.80 (Haywood et al., 2003). Thus, even within a single campaign, past work has shown a sizable range in BB aerosol properties. It should be noted that the SAFARI over-ocean flights were conducted within a more southern region (generally 15°-25°S) than the heart of the seasonal aerosol plume typically described as extending approximately 0°-15°S (Zuidema et al., 2016). The ORACLES

sampling area spans both these latitude ranges (0°-25°S), but frequently sampled westward of the SAFARI region. SAFARI also made many measurements over the continent, closer to biomass burning sources, whereas the ORACLES measurements were made entirely over the ocean. It is therefore likely that the SAFARI measurements were of generally younger aerosol than the ORACLES measurements, including the aerosol identified as 'aged' within Haywood et al. (2003).

An AERONET-based climatology by Dubovik et al. (2002) indicated that the African BB site (Zambia, 15°15'S, 23°09'E)

had the lowest SSA (i.e. strongest absorption) and the strongest SSA spectral dependence (i.e., steepest slope, $SSA_{440nm}$=0.88 to $SSA_{1020nm}$=0.78) among the four geographical BB regions considered. This was attributed to the greater flaming versus smoldering characteristics of these fires compared with other regions. Using data from the same long-term AERONET site at Mongu, Zambia, Eck et al. (2013) found a seasonal progression of increasing full-column SSA (decreasing relative absorption) over the July-to-November burning season based on data from 1997 to 2005. SSA increased from $SSA_{440nm}$=0.84 in July to

$SSA_{440nm}$=0.93 in November (Figure 1). This same pattern was observed in a study of near-surface in situ SSA measurements





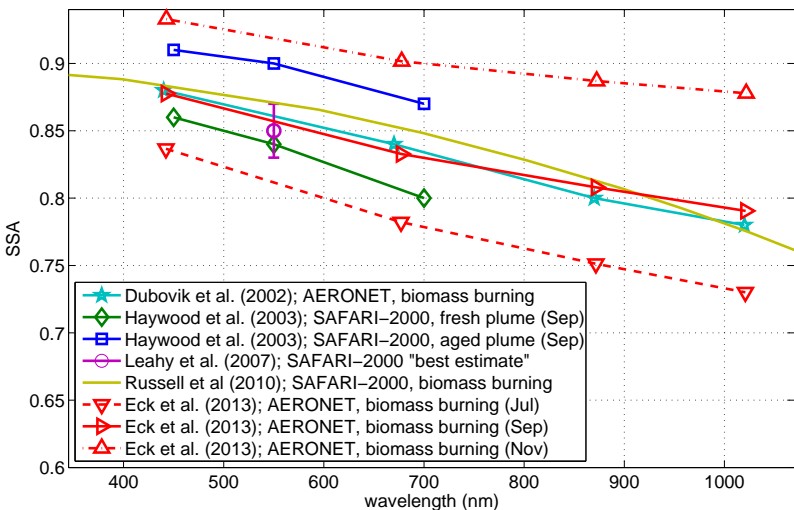

**Figure 1.** Summary of average SEA/African BB SSA as reported in previous studies as described in the text. The spatial and methodological variability can be noted from the Haywood et al. (2003) vs Russell et al. (2010) SAFARI 2000 results, and the seasonal changes in SSA are seen in the Eck et al. (2013) monthly averages from the Zambian AERONET site.

at Ascension Island, showing monthly mean $SSA_{529nm}$ increasing from 0.78 in August 2016 to 0.83 in October 2016 (Zuidema et al., 2018). We note this site is substantially westward (downwind) of the majority of ORACLES flights, and thus may represent more aged aerosol. Eck et al. (2013) also reported an average September SSA similar to that reported by Dubovik et al. (2002) for Zambia: 0.88, 0.83, 0.81, and 0.79 at 440, 675, 870, and 1020nm, respectively. They also showed that within

5 a given month, a south-to-north increase in $SSA_{388}$ is observed, as derived from the satellite-based OMI (Ozone Monitoring Instrument) retrievals. This was hypothesized to be due to shifts in fuel type over the BB season due to both anthropogenic factors (i.e. timing/practices of agricultural burning) and environmental factors (i.e. relating to moisture variability of potential fuel types throughout the season). Variations in atmospheric humidity have also been shown to affect aerosol optical properties, including both aerosol scattering and absorption (e.g., Langridge et al., 2011; Lack and Cappa, 2010). Figure 1 summarizes the

10 previous work in this region, and indicates that the SSA over southern Africa increases through the burning season (monthly averages from Eck et al., 2013), possibly due to changes in biomass burning fuel composition, and also that SSA may increase with distance from fire, i.e. as the aerosol ages (fresh versus aged plume of Haywood et al., 2003). However, we emphasize that the "aging" time scales observed in SAFARI-2000 were much shorter (e.g. ∼5 hours in Abel et al. (2003); within a few days in Haywood et al. (2003)) than those seen in ORACLES (∼2-15 days, e.g. Dobracki et al., 2019). This may account for

the opposite sign (higher SSA for younger aerosol) shown in the latter work. Thus, it is clear that multiple physical factors may be responsible for variability in aerosol optical properties.

Further discussion of previous studies of SEA aerosol in the context of the ORACLES results is found in Section 4.



## 1.2 Impacts of SSA on aerosol radiative effects

Accurate representation of scattering versus absorption within biomass burning aerosol scenes has implications in subsequent calculations of aerosol radiative effects over the SEA. The range of SSA values shown in Figure 1 encompasses the possibility of both positive (smaller values of SSA) or negative (higher values of SSA) net direct radiative effect when this aerosol is

above the partly-cloudy skies of the SEA. The specific value of SSA where the radiative effect changes sign will depend on the cloud cover fraction; for example, in Chand et al. (2009) the aerosol direct effect was positive for a mid-visible SSA of 0.85 as long as cloud fraction was greater than 40%. Wilcox (2012) found that for a perturbation to SSA of $\pm 0.03$, the local direct aerosol radiative effect was changed by 10-20 W/m$^2$, though this also depends on AOD and cloud albedo conditions. A perturbation of this magnitude encompasses only half the range between the lowest and highest September SSA values shown

in Figure 1. As described above, some of this range may reflect seasonal variations in SSA, but even for September alone, the previously measured values of $SSA_{550nm}$ spanned 0.84-0.90. An open question is to what degree the range in previous observations represents real variability in the SSA of smoke in this region or whether a significant part of the range is due to differences in measurement techniques and in measurement conditions; for example, the AERONET and flux radiometry retrievals are for ambient-RH aerosol, whereas the in-situ (SAFARI) measurements are of dry/low-RH aerosol. We endeavor

to explore these questions in the following sections.

## 1.3 ORACLES overview

The overarching goal of ORACLES is to make high-quality airborne observations of aerosols and clouds in the SEA to gain a better understanding of the complex processes (direct, indirect, and semi-direct) by which BB aerosols, notable for their strong absorption of solar energy, affect radiation both directly and through their impacts on clouds (Zuidema et al., 2016). The project

included three field deployments of approximately one month each: September 2016 based out of Walvis Bay, Namibia; August 2017 based out of São Tomé, São Tomé and Príncipe; and October 2018 again based out of São Tomé. The ORACLES study area time zone spans UTC and UTC+1. In the current paper we focus on the first field deployment of ORACLES in September of 2016 out of Walvis Bay, Namibia. This deployment included two NASA aircraft: a P-3 for full atmospheric profiling and low/mid-level in situ sampling, and a high-altitude ER-2 for remote sensing observations. The P-3 aircraft was flown with

a suite of in situ and remote-sensing aerosol, cloud, radiation, and meteorological instruments, while the ER-2 carried only remote-sensing instrumentation. Data were collected over 15 P-3 and 12 ER-2 flights, each 7-9 hours in duration. The 2017 and 2018 deployments included the P-3 only. Hence, the simultaneous deployment of both the P-3 and the ER-2 in the 2016 deployment created a unique testbed for evaluating remote sensing retrievals of aerosol and cloud properties from a variety of instruments that have potential for future space flight.

Of the in situ and remote-sensing instruments included in ORACLES, eight teams (including the complementary AERONET sites, several of which were established to coordinate with the ORACLES deployments) observe or derive aerosol absorption (either locally or as column AAOD) and the related SSA parameter (Table 1). All considered remote-sensing instruments report an AOD product, with the exception of the Solar Spectral Flux Radiometer (SSFR), which uses AODs from the Spectrometers





for Sky-Scanning, Sun-Tracking Atmospheric Research (4STAR) as input (each of these instruments is described below). Remote-sensing SSA are reported as column-integrated values; for in-situ measurements, the SSA presented is the extinction-weighted profile-average value (i.e. SSA calculated at each altitude and weighted by the profile of extinction; Section 2.1.3). Unless otherwise noted, reported Ångström exponents are calculated using a logarithmic fit of the AOD versus wavelength

5  using all available wavelengths between 440 and 675nm (inclusive), to have the most comparable quantity between instruments and to reduce uncertainty compared with a simple 2-wavelength calculation of AE. While previous studies have examined the agreement between the retrievals of a few of these instruments at a time (e.g., Sedlacek and Lee, 2007; Leahy et al., 2007; Knobelspiesse et al., 2011), an intercomparison including so many methods within one campaign has not previously been performed, due to the logistics of assembling such a comprehensive suite of instruments including several newly-developed

10  algorithms. Specific instrument details are given in the following section.

## 2 Methods

### 2.1 Instruments and data

In this section we offer descriptions of the instruments and data included in this paper, as summarized in Table 1.



**Table 1.** Overview of ORACLES instruments as used to quantify SSA during the ORACLES campaigns, and the method used. Not all measurement approaches are included in this paper; some are still in process and will be presented in future publications. Instruments/methods presented in this paper are indicated in bold. Full archival data citations are given under Acknowledgments.

| Instrument | Platform | Method (absorption parameter) | Wavelengths | Required conditions |
|---|---|---|---|---|
| 4STAR (+SSFR) | P-3 | **AERONET-like sky scans (AAOD)**[1,2,13] | hyperspectral; evaluated using AOD and radiances at 400, 500, 675, 870, 995nm | instrument below aerosol; level flight leg; cloud-free above |
| SSFR (+4STAR) | P-3 | **Profiles of spectral irradiance and AOD used with a radiative transfer model (RTM)** [3,4] | moderate spectral resolution; evaluated at 355, 380, 452, 470, 501, 520, 530, 532, 550, 620, 660nm | radiation wall/square spirals above/below aerosol layer |
| PSAP + Neph | P-3 | **in situ nephelometer (scattering) + PSAP (absorption)**[5,6] | Neph: 450, 550, 700nm PSAP: 470, 530, 660nm | inside aerosol layer |
| PTI + Neph | P-3 | **in situ nephelometer (scattering) + PTI (absorption)**[7] | Neph: 450, 550, 700nm PTI: 532nm | inside aerosol layer; 30s averages on PTI |
| RSP | P-3 +ER-2 | polarized reflectances input into RTM[8] <br><br> neural network[9] <br><br> **MAPP algorithm: optimal estimation method using multi-angle, multi-channel total & polarized reflectances (bimodal column absorption/AAOD)**[10] | I, Q, and U Stokes components in 410*,469*, 555*, 670*, 864*, 960, 1594*, 1880, and 2264*nm (* denotes window channels used in the retrieval of aerosol properties) | instrument above aerosol |
| RSP + HSRL-2 | ER-2 | MAPP algorithm adapted to include HSRL-2 observed backscatter & extinction profiles (vertically-resolved absorption)[11] | RSP window channels +HSRL-2 channels at 355, 532, 1064nm | instrument above aerosol |
| AirMSPI | ER-2 | **multi-angle, multi-spectral polarized radiance fitted by a RTM to derive above-cloud aerosol optical and microphysical properties which are used to calculate AAOD**[12] | Stokes component I in bands centered at 355, 380, 445, 470, 555, 660, 865nm and Stokes components Q and U in 470, 660, 865nm bands | instrument above aerosol; operated in sweep view mode |
| AERONET | ground | sun radiance + multiangular sky radiances into RTM (AAOD)[13] | 440, 675, 870, 1020nm | cloud free above instrument |

[1]Redemann et al. (2014) [2]4STAR codes (see Acknowledgments) [3]Bergstrom et al. (2010) [4]Cochrane et al. (in prep) [5]Anderson and Ogren (1998) [6]Virkkula (2010) [7]Sedlacek and Lee (2007) [8]Knobelspiesse et al. (2011) [9]Knobelspiesse et al. (in development) [10]Stamnes et al. (2018) [11]Liu et al. (in development) [12]Xu et al. (2018) [13]Dubovik and King (2000)



### 2.1.1 4STAR

The Spectrometer for Sky-Scanning Sun-Tracking Atmospheric Research (4STAR) is an airborne hyperspectral (350-1700 nm) sun photometer which can make direct-beam (sun-tracking mode) measurements for retrieval of column AOD and trace gases (Dunagan et al., 2013; Shinozuka et al., 2013; Segal-Rosenheimer et al., 2014) or below-cloud measurements of transmittance

for derivation of cloud optical properties (zenith mode). Under certain level-flying conditions, 4STAR can also perform an AERONET-like sky scans in either the principal-plane or almucantar (sky scanning mode), which provide the data used here. The 4STAR sky scans are processed using a modified version of the Version 2 AERONET retrieval algorithm described in Dubovik and King (2000), which retrieves aerosol size distributions, refractive indices, SSA, and AAOD, among other parameters. All scans are run through the algorithm with the minimum scattering angle set to 3 degrees to avoid stray light from

the sun entering the 4STAR optical aperture. Scene (i.e., surface plus atmosphere) albedo is provided by SSFR measurements (described below). A notable modification of the 4STAR retrievals compared with AERONET retrievals is in the input wavelengths. While AERONET uses radiances measured at specific and discrete wavelengths (440, 675, 870, and 1020nm), with the hyperspectral 4STAR we are able to use AODs and sky radiances measured at a different (or larger) selection of wavelengths. Due to suspected stray light contamination within the 4STAR spectrometer around 440nm, a particular sensitivity to the 440nm

channel was observed, which in some cases resulted in an anomalously low SSA (high AAOD) at shorter wavelengths compared with retrievals run without 440nm. To avoid this issue, the results presented in this paper use a modified set of inputs at wavelengths of 400, 500, 675, 870, and 995nm, with 400 and 500nm replacing 440nm. Note the longest wavelength of 995nm replaces the AERONET 1020nm due to the wavelength limits of the 4STAR visible spectrometer.

    4STAR executed a total of 174 sky scans in ORACLES-2016, of which 38% (66) met the following quality control (QC)

criteria (adapted from the AERONET QC available at https://aeronet.gsfc.nasa.gov/new_web/PDF/AERONETcriteria_final1. pdf):

- AOD(400nm) > 0.4

- altitude variation during scan < 50 m

- total residual sky radiance error ($|$measured-fit$|$) < 10% at all wavelengths

- measured scattering angle from the minimum of 3 degrees up to a maximum of at least 50 degrees (i.e. SZA > 25 degrees) (primarily relevant in almucantar scans)

- manual inspection of the retrieval output for reasonable residual sky radiance error as a function of scattering angle (i.e. uniform aerosol conditions, no cloud contamination)

    Note that, unlike in the AERONET archive, we consider principal plane as well as almucantar scans when the above criteria

are met. We do this because due to the timing of ORACLES flights, 4STAR sky scans were largely near solar noon, limiting the angular range available in almucantar scans. In addition to the scans meeting the above QC measures, another 16 (9% of the scans) were included based on manual QC inspection. This generally involved cases with AOD between 0.2 and 0.4. We retain





these lower-AOD, manually QC'd scans to explore the reliability of the retrievals under conditions of lower aerosol loading. This QC procedure retained 82 sky scans in total (47% of all scans) which produced credible retrievals. In the present study, we focus on a further subset of 75 lower-altitude ($< 3$km), QC-screened sky scans. This is because our current interest is in retrievals of aerosol properties through the entire aerosol plume, and thus the low-altitude sky scans are most comparable to the

other instruments presented here. 49 of the QC-passed scans corresponded to valid in situ data and are used in the aggregate comparison figures; an additional 26 did not correspond in space and time to the other instruments, but are included in the broader analysis of Section 4. The available sky scan retrievals and their co-location with other instruments are summarized in Table 2.

The 4STAR uncertainties presented here are quantified by a sensitivity test based on AOD and radiance uncertainties. AOD

uncertainties used are the archived wavelength-dependent uncertainties (LeBlanc et al., 2019), and uncertainties in the sky radiances have been quantified through laboratory calibration using a NIST-traceable 12-lamp 36-inch integrating sphere (Brown et al., 2005). AOD uncertainties are dependent on wavelength, time, and solar zenith angle (geometrical air mass factor), as well as potential window contamination in some cases. These values were typically between 0.01 and 0.02, ranging from a low of 0.008 to a high of 0.037 in an extreme case. Radiance uncertainties are wavelength-dependent, but are constant over

the entire campaign, ranging between 1.0% and 1.2% for 470-995 nm. To test the impact of these two types of errors, the sky scan inversion code is run separately for an addition or subtraction case for each of these two parameters (i.e. four cases), and the result is added in quadrature for each of the upper and lower bounds. Note that an increase in AOD (without perturbing radiances) results in a lower SSA and higher AAOD, while an increase in radiance (without perturbing AOD) results in a higher SSA and lower AAOD. Uncertainties in SSA are dominated by the AOD terms, with smaller contributions from the uncertainty

in the measured sky radiances. Other sources of uncertainty are not explicitly quantified in the present work.

### 2.1.2 AirMSPI

The Airborne Multi-angle SpectroPolarimeter Imager (AirMSPI) is an imaging polarimeter which flew on the ER-2 aircraft in ORACLES-2016 (Diner et al., 2013). The instrument has been flying aboard the NASA ER-2 high altitude aircraft since October 2010 and has two operational modes: step-and-stare view mode and sweep mode, with 25-meter and 10-meter spatial

resolution, respectively. The sweep view mode was adopted for cloud and above-cloud aerosol observations during the ORACLES field campaign. The AirMSPI data presented here are from a coupled stratocumulus cloud and above-cloud aerosol retrieval based on an optimization approach (Xu et al., 2018). The retrieval is run by fitting polarized radiance in a wide scattering angular range (e.g. from $\sim 90°$ to $180°$) at three spectral bands centered at 470, 660, and 865 nm. The retrieved above-cloud aerosol properties include refractive index, size distributions, and aerosol total volume concentration. The re-

trieved cloud properties include cloud-top droplet size distribution, cloud-top height, and cloud optical thickness (cloud optical thickness is derived by fitting the radiance in the three polarimetric bands). Non-spherical particles are not accounted for in the current retrievals. The column effective AOD and SSA are calculated using Mie theory. Retrieval uncertainties are reported at the polarimetric wavelengths, and are determined by propagating the instrument errors into the retrieval uncertainties. For example, to get the retrieval uncertainty for above-cloud AOD and SSA, the fitting residual plus instrument bias are multiplied



### 2.1.3 HiGEAR (PSAP+Nephelometer)

The Hawaii Group for Environmental Aerosol Research (HiGEAR) operated several in-situ instruments on the P-3. Total and
sub-micrometer aerosol light scattering coefficients ($\sigma_{\mathrm{scat}}$) were measured onboard the aircraft using two TSI model 3563
3-wavelength nephelometers (at 450, 550, and 700 nm) corrected according to Anderson and Ogren (1998). In addition to the
TSI nephelometers used in the present work, two single wavelength nephelometers (at 550 nm, Radiance Research, M903)
were operated in parallel to study the increase in light scattering as function of relative humidity (RH). The humidified M903
nephelometer was operated near 80% RH while the dry unit was maintained below 40% (Howell et al., 2006). Discussion of
the impacts of aerosol humidification in the context of comparison with remote-sensing retrievals at ambient RH is found in
Section 4.2.

Light absorption coefficients ($\sigma_{\mathrm{abs}}$) at 470, 530, and 660 nm were measured using two Radiance Research particle soot
absorption photometers (PSAPs). The humidity within the PSAP was not explicitly controlled, but the PSAP optical block
was heated to approximately 50°C to reduce artifacts which would result from a changing RH; this had the effect of reducing
relative humidity in this instrument to much lower than the 40% within the nephelometers. The PSAP absorption corrections
were performed according to an updated algorithm (Virkkula, 2010). Instrumental noise levels are 0.5 Mm$^{-1}$ for a 240–300 s
sample average, comparable to values reported previously (Anderson et al., 2003; McNaughton et al., 2011). In this paper, we
primarily present results calculated with the wavelength-averaged (as opposed to the wavelength-specific) correction factors
presented in Virkkula (2010). Further discussion of this decision and the differences between the two corrections are shown in
Appendix A.

SSA was calculated using the measured PSAP absorption combined with dried (RH<40%) TSI nephelometer scattering
interpolated to PSAP wavelengths. In the comparison cases (i.e. column values), SSA is the extinction-weighted (extinction =
scattering + absorption) profile average according to the following procedure. To reduce noise, the reported 1-second scattering
and absorption data, corrected to ambient temperature and pressure, were first averaged to 30s box-car averages. Time-averaged
data are then filtered to reject cases where $\sigma_{\mathrm{scat},530\mathrm{nm},30s} < 10\,\mathrm{Mm}^{-1}$ to assure an adequate signal-to-noise ratio, Data are also
discarded if >20% of the archived 1s SSAs are undefined over the averaging period, to account for manual quality flagging
of SSA due to, e.g., calibration periods. Applying different averaging times (10-60 s averages) did not result in appreciably
different SSA and Ångström exponents for the column-average. SSA is then calculated as SSA=$\sigma_{\mathrm{scat}}/(\sigma_{\mathrm{scat}} + \sigma_{\mathrm{abs}})$, and
then arithmetically weighted by its extinction in computing a profile average. In the AOD proxy shown in Section 3.2, due to
the vertical integration involved, scattering and absorption data were instead averaged into equal 100-m vertical bins (approxi-
mately 15 s of flight time) and integrated over the full profile; in this specific case, only profiles with altitudes spanning at least
1.6 km to 5.1 km are considered (Table 2).



Unless otherwise specified, the "in situ" data reported in this paper are from the PSAP+Nephelometer combination, with Virkkula wavelength-averaged corrections applied.

### 2.1.4   PTI + Nephelometer

The second in situ measurement of aerosol absorption uses data from the airborne photothermal interferometer (PTI). The PTI measures aerosol light absorption by combining photothermal spectroscopy and laser interferometry (Sedlacek, 2006; Sedlacek and Lee, 2007). The hallmark of the PTI, and other photothermal-based techniques, is a complete immunity to light scattering. The unit deployed during ORACLES-2016 operated at a wavelength of 532 nm. While the baseline noise on the ground was $\sim$1.5 Mm$^{-1}$ for a 1s integration, the noise floor during in flight increased to $\sim$20 Mm$^{-1}$, due to platform (mechanical) vibrations. This shortcoming was addressed in a revision of this instrument. Due to these mechanical limitations, the PTI data are archived as 30s averages, with data available for some periods and for a subset of P-3 flights only. Laser overheating further limited the operation of this instrument to in-plume transects. In the multi-instrument comparison case studies presented here, the extinction-weighted SSA and absorption (AAE) are calculated from PTI absorption for data within the time period of the P-3 comparison case (profile and in-plume leg). The whiskers shown on PTI absorption in the comparison figures show the 10th-90th percentiles over the comparison window (i.e. the variability during a comparison window). A direct comparison of the two in-situ instruments (PTI and PSAP) is shown in Section 3.1.

### 2.1.5   RSP

The NASA GISS Research Scanning Polarimeter (RSP) is a multi-angle, multi-spectral polarimeter aboard the ER-2 that measures the Stokes parameters I, Q, and U at $\sim$150 angles between $\pm \sim 60°$ in the along-track direction, in 9 spectral channels centered at 410*, 469*, 555*, 670*, 864*, 960, 1594*, 1880, and 2264* nm (Cairns et al., 1999). The seven channels denoted by an asterisk have negligible or weak and correctable water vapor absorption and were used in the microphysical aerosol properties from polarimetry (MAPP) retrieval. The RSP MAPP algorithm (Stamnes et al., 2018) was adapted for ORACLES observations of aerosols above water by incorporating into the retrieval an aerosol profile consisting of two layers (a top layer of fine mode aerosol located at 2.25 – 5.5 km, and a base layer of coarse mode (sea salt) aerosol located at 0 – 1 km) as approximately identified by the High Spectral Resolution Lidar (HSRL-2), which also on the ER-2 aircraft. The aerosols are modeled as a bimodal population of spherical fine- and coarse-mode aerosols, with each mode defined by a lognormal size distribution. The fine mode aerosol effective radius, effective variance, and complex refractive index are then retrieved. The coarse mode aerosol is assumed to consist of nonabsorbing spherical particles with complex refractive index equal to that of water, except the real part was multiplied by a factor of 1.01. The maximum allowed windspeed for the one-dimensional Cox-Munk ocean was increased to 12 m/s, to allow retrieval of high windspeeds consistent with MERRA2 profiles.

Since for RSP the fine and coarse modes are retrieved separately, the total AOD is thus the sum of the fine plus coarse mode optical depths. Due to the assumption that coarse mode aerosol is nonabsorbing sea salt, the AAOD and Ångström exponent values are provided for the fine mode only, to allow for more direct comparisons to the smoke properties retrieved by the remote sensors above clouds.



### 2.1.6 SSFR

The Solar Spectral Flux Radiometer (SSFR) is a moderate resolution radiative flux (irradiance) spectrometer covering the wavelength range from 350 to 2100 nm (Pilewskie et al., 2003; Schmidt and Pilewskie, 2012). The downwelling (zenith) and upwelling (nadir) solar radiation is collected by light collectors mounted to the skin of the aircraft. In the past, aerosol absorption, SSA, and asymmetry parameter were derived from irradiance pairs collected along collocated horizontal legs above and below the layer (Schmidt et al., 2010). For ORACLES, this approach was impractical because of the underlying albedo variability in the presence of clouds. The alternative is to measure the irradiances in a vertical profile, realized as a spiral. This was not an option for previous experiments where the zenith light collector was fix-mounted to the aircraft, introducing uncertainties due to the changing aircraft attitude that cannot be corrected for after the fact. Specifically, for ORACLES, an Active Leveling Platform (ALP) was built for the zenith light collector on the P-3 aircraft. By controlling its angular position, SSFR is able to obtain zenith (downwelling) irradiance measurements throughout the vertical profile if the spirals include short ($\sim$9-30s) straight segments (typically offset by 90° in heading). This spiral profile maneuver with short straight legs is referred to as a square spiral. The nadir (upwelling) irradiance measurements are affected by the underlying cloud and its variability, as well as by the aerosol between the cloud top and the nadir light collector. To separate the aerosol signal from that of clouds, SSFR uses the upwelling irradiance at 1.6 $\mu$m where the signal is dominated by clouds and filter the data such that only points within one standard deviation of the mean are included for final processing. The impact of the aerosol layer on the downwelling and upwelling irradiance is then quantified throughout the spiral by plotting spectral irradiance profiles with 4STAR-reported above-aircraft AOD at 532nm as the vertical coordinate. A linear fit is performed on both irradiance components with respect to the AOD, and absorption is then derived as the change in net irradiance over the AOD differential over the full spiral. Since this approach uses data throughout the vertical profile, it is a more robust and accurate method than obtaining it just from the irradiance pairs above and below the layer as in a radiation wall. More importantly for measurements above clouds, this method minimizes the impact of cloud variability on the sampling of upwelling irradiances through the filtering approach described above. Any deviations from a linear relationship between AOD and irradiances are attributable to changes in the underlying cloud albedo. The filtering technique allows separation of these influences from those originating from the aerosol layer. SSA is retrieved with an algorithm that iteratively changes SSA and asymmetry parameter until the modeled irradiance profiles (based on 4STAR AOD and SSFR-derived cloud albedo) match the measurements (Schmidt et al., 2010). The SSA retrieval is done independently for each wavelength, without applying spectral smoothness constraints. Uncertainties in SSFR SSA as reported here reflect the 1-sigma uncertainty as calculated from the probability of the SSA and asymmetry parameter pair within the retrieval. A description of the algorithm and the uncertainty analysis may be found in (Cochrane et al., in prep).

### 2.2 Instrument intercomparison conditions

Comparison cases were selected based on the available instrument data and the flight path for a given day. Identified cases included at least two of the following conditions to facilitate comparison between instruments:



1. Operational nephelometer plus PSAP and/or PTI in situ data during an aircraft profile (either a ramp or a square spiral as described in Section 2.1.6);

2. ≥1 sky scan (4STAR) at or below the bottom of the plume (i.e. measuring the total-column aerosol properties);

3. a square-spiral profile through the full column (SSFR);

4. an ER-2 overpass of the P-3 location (RSP and/or AirMSPI).

Due to the different flight patterns necessary for the different instruments to measure/retrieve aerosol properties, this yielded 24 potential comparison case studies over 12 (out of 14) flight days (Figure 2). Within this set of cases, temporal separation between measurements from different instruments varies between 10 minutes and 2 hours; the inter-measurement spatial spread was within approximately 1 degree in either latitude or longitude (ideally 100km or less, but up to 130km in select cases). Note
that not all instruments were available for each comparison (Table 2). In selecting comparison periods spatial coincidence was given priority over temporal coincidence. All comparison cases included at least one 4STAR sky scan; this was reduced to 20 cases when we required QC'd sky scans below 3km (considered to be below the bulk of the aerosol plume), 19 of which were coincident with an in-situ profile. Of the 24 cases, 14 had full profiles and 9 had partial profiles only, for a total of 23 of the 24 with in situ observations of SSA.

Below we focus in on two specific case studies, on 12 and 20 September 2016. The case on September 12 (blue star in Figure 2) was a mostly cloudless scene and included RSP retrievals from an ER-2 overpass. For the case on September 20 (orange star in Figure 2), valid SSFR retrievals were run using data from a P-3 square spiral maneuver through a cloudy scene overlaid with substantial aerosol loading. Each of the twelve ER-2 flights (including transits to/from Namibia) yielded AirMSPI ACA retrievals somewhere over the SEA; nine of the P-3 comparison cases included ER-2 overpasses, and each of these had at least
one AirMSPI ACA retrieval co-located with P-3 observations. Three of these cases (including the case from 20 September discussed in Section 3) had retrievals with the highest confidence (labeled "primary" in Table 2); retrievals were run for an additional 6 cases, which are included in Section 4 to allow for broader comparison. However, these second-tier retrievals have somewhat increased potential for retrieval biases, due to small scattering angle coverage and/or broken cloud conditions. Successful above-cloud aerosol retrievals for data from the RSP have been processed for one of the cases thus far (the case
study on 12 September). While the present work is thus limited to a subset of the 24 cases, future comparative analysis may be able to expand the number of cases to incorporate potential newly-available data (Table 1).

It is important to note that due to the different instrument methodologies, exact spatiotemporally coincident measurements are not possible, if for no reason other than the different viewing geometries alone; the comparisons presented here are chosen for their potential to obtain measurements of reasonably similar aerosol properties from different perspectives (e.g. below-
versus above-aerosol remote sensing, and remotely sensed versus in situ; Figure 3). In our analysis, we first present the two individual case studies (Section 3.2) before discussing results from the aggregation of all coincident measurements from three of the instruments (Section 3.3).



**Table 2.** Summary of data available for instruments with multiple comparison cases (i.e. from 4STAR, AirMSPI, and in situ instrumentation), for campaign-wide statistics (top row; Section 4) and for comparison with other instruments (lower rows; Section 3).

|  | 4STAR (P-3) | AirMSPI (ER-2) | in situ (P-3): full profile | in situ: partial profiles only |
|---|---|---|---|---|
| # valid retrievals | 75 (below 3km) of 82 (total) | 134 (primary) +68 (additional) | continuous | continuous |
| # cases total | 19 | 9 | 14 | 9 |
| vs 4STAR | — | 9 | 11 | 14 |
| vs AirMSPI | 9 | — | 6 | 3 |
| vs in situ | 19 | 9 | — | — |

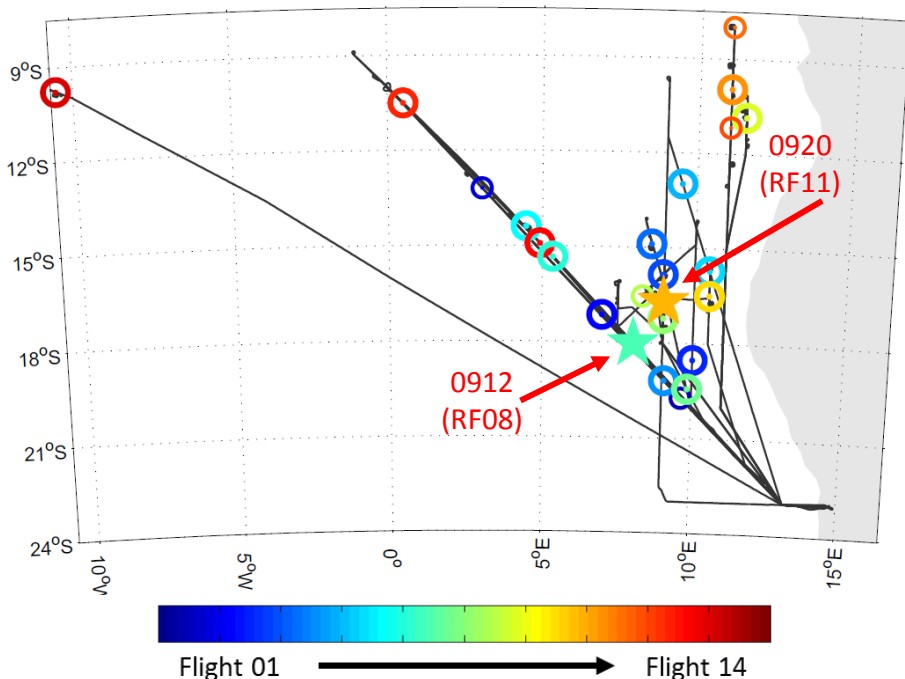

**Figure 2.** Summary of data available for instruments with multiple comparison cases (i.e. from 4STAR, AirMSPI, and in situ instrumenta-tion), for campaign-wide statistics (top row; Section 4) and for comparison with other instruments (lower rows; Section 3).



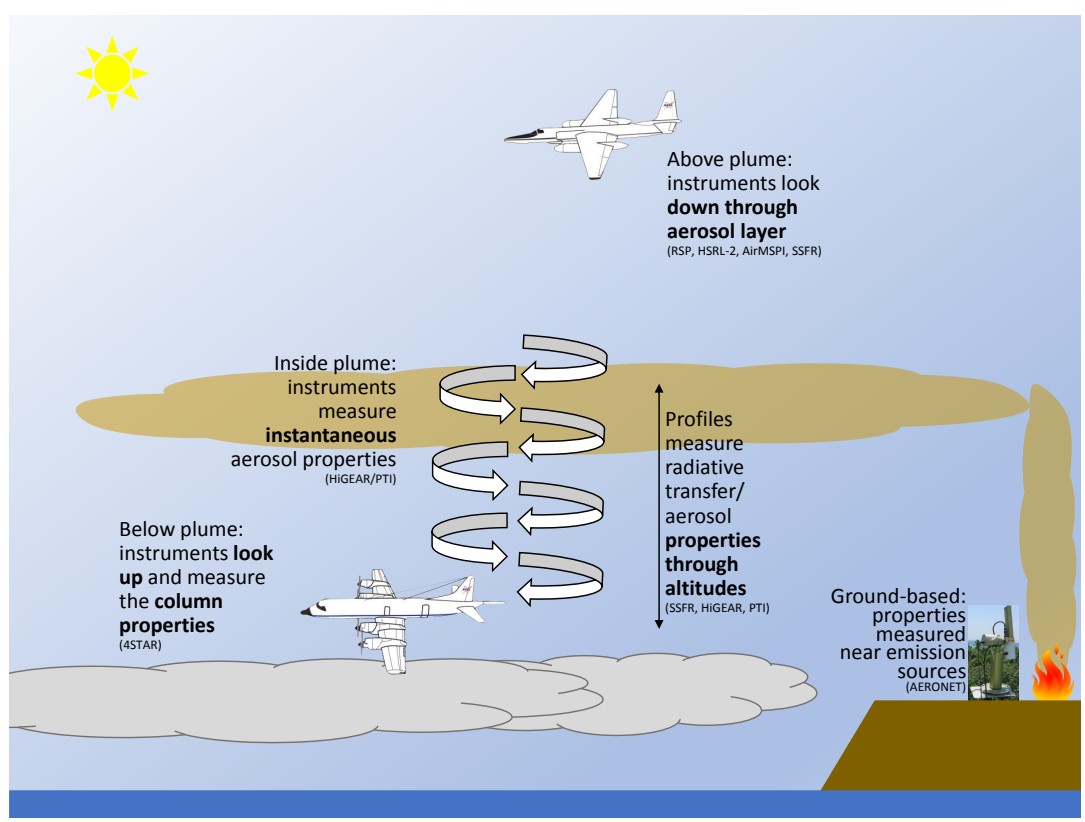

**Figure 3.** Schematic illustrating the different successive measurement orientations necessary for a comparison between different instruments on different aircraft.

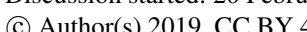



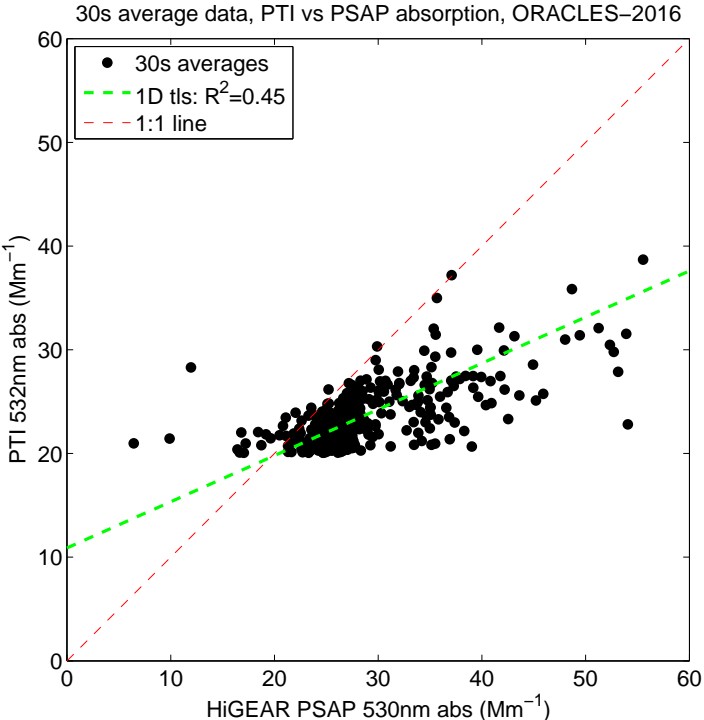

**Figure 4.** Comparison between PTI vs PSAP aerosol absorption coefficient (at 532 and 530nm, respectively) for all available flight data. Lines show a 1:1 relationship (red dashed) and a total-least-squares fit (green dashed) through the data.

## 3 Results

### 3.1 Comparison of absorption from in situ instruments

As this paper discusses SSA derived using two different in-situ methods, we first briefly discuss the two in-situ absorption instruments as they compare to one another. Figure 4 shows 30-second averages of absorption from the PTI and from the PSAP

5  for all available coincident data for the 5 flights with the PTI operational (flights on September 10, 12, 14, 20, and 24). The effect of the high noise floor applied to the reported PTI values is evident in the data variability, and the PTI absorption is seen to be generally lower than PSAP absorption over all days, consequently with a smaller dynamic range: 5th-95th percentiles of 20.3 to 28.8 Mm$^{-1}$ (median 22.8 Mm$^{-1}$) for PTI compared with 21.8 to 40.9 Mm$^{-1}$ (median 26.6 Mm$^{-1}$) for the coincident PSAP measurements. The generally lower PTI absorption could be due to several factors, including: an artificially-low-biased

10  PTI signal driven by latent heat of evaporation under high-humidity aerosol conditions or an artificially-high-biased PSAP signal due to multiple scattering from the filter-based measurement. Improvements in the PTI instrument design for the 2018 deployment will likely facilitate better comparative analysis between the two measurement techniques.



## 3.2 Multi-instrument case studies

We next show two specific case studies, from flights on 12 September and 20 September 2016, where 6 of the methods can be compared.

### 3.2.1 Case study: 12 September 2016 (broken cloud/clear-sky)

The P-3 flight on 12 September was a radiation-targeted flight of opportunity, with two potential comparison cases identified. Both cases are included in the analysis in Section 3.3, but in this section we focus on the second case, at approximately 18.0°S, 8.0°E, between 13:44 and 14:34 UTC (Figure 2). This case starts with two consecutive 4STAR sky scans at approximately 1 km altitude (above-cloud-level) over a broken cloud scene of albedo of 0.1, followed by a short descent and two scans at 80m, which is below typical cloud level but was, in this case, in a cloudless area (scene albedo approximately 0.05). The P-3

then flew a ramped ascent from 80m to 5.8 km, ending above the top of the aerosol plume. Measured relative humidity (RH) throughout the plume increased from near 0% below the plume level (up to 2.5km) to a maximum of around 50% at plume top (5km), or generally below the RH=40% "dry" threshold for the in situ instrumentation. This corresponds to a roughly constant water vapor mixing ratio: around 5000ppmv through the plume. The first pair of sky scans (at 1km altitude) are separated from the second pair (at 80m, at the base of the ramp) by approximately 20 minutes and 130 km. While this is slightly outside our

desired spatial constraints, we believe it instructive to retain the two above-boundary-layer sky scans in examining this case, first to facilitate better comparison with RSP (which retrieves above-cloud fine-mode aerosol separately) and second because the bulk of the ORACLES-2016 data consists of plume-only aerosol without boundary-layer influence, and thus inclusion of these scans allows for better contextualization of this specific case. The ER-2 overpass of the scene, occurring during the P-3's ascending ramp, resulted in 20 RSP retrievals between 14:17 and 14:21 UTC. There were no AirMSPI retrievals during this

period.

  Figure 5 shows the SSA for each of the available instruments for this case. The most notable feature is in the 4STAR sky scans; while the retrievals agree quite well at short wavelengths (within ∼0.01 and well within 4STAR's uncertainty range), at the longer wavelengths the two sets of scans diverge markedly. It may be noted that the uncertainty increases as well at these longer wavelengths, and there is overlap in the two sets of uncertainty estimates. The two sky scans performed near the

surface (at 80m altitude), immediately before the start of the ascending ramp, agree well with one another and give a higher long-wavelength SSA than the two scans performed above the boundary layer (at 1km altitude). However, the 1km scans are more representative of the typical SSA spectral shape observed from 4STAR for the ORACLES-2016 campaign as a whole (i.e., monotonically decreasing for wavelengths ≥500nm). As the majority of the 4STAR sky scans were taken above the boundary layer, the increase in SSA at longer wavelengths as seen in the two 80m scans likely reflects contribution from sea

salt in the boundary layer, combined with the overlying biomass burning aerosol. In situ measurements report $\sigma_{\text{scat},530\text{nm}}$ of around 30Mm$^{-1}$ between 80 and 600m with negligible absorption in these altitudes, suggesting purely scattering sea salt. This is additionally corroborated by the HSRL-2 retrievals of aerosol type, which identified marine aerosol below the smoke plume around this time. The AERONET climatology of aerosol from "desert dust and oceanic" sites (Bahrain/Persian Gulf, Cabo





Verde, and Lanai, HI sites) by Dubovik et al. (2002) shows high, largely spectrally-flat or slightly increasing SSA between 440nm and 1020nm. Indeed, LeBlanc et al. (2019) also found lower extinction Ångström exponents for "full column" (i.e. below cloud level) 4STAR AOD measurements than for those taken above the boundary layer, which is consistent with the differences seen in this case (Figure 6).

In situ measurements show a similar spectral SSA with wavelength compared with the other two instruments. We note that if we apply the wavelength-specific, rather than wavelength-invariant, Virkkula corrections to the PSAP data, the PSAP+Neph SSA shows a different spectral shape, with a small maximum in SSA at 530nm (Appendix A). It is also important to note that the in situ measurements will exclude a significant portion of the coarse-mode aerosol due to poor inlet passing efficiency of larger aerosol particles (a 50% size cut around $\sim$4 microns), which means that nominally there are twice as many 4$\mu$m

particles observed by remote sensing instruments compared with in situ, and perhaps 10 times as many at larger sizes (10 - 20 $\mu$m), and thus even "full column" in situ values may be missing larger aerosol. Thus, the SSA and AEs derived from the in situ instruments will have a greater contribution from the biomass burning aerosol than will the values retrieved from 4STAR, (and in the later comparison case, AirMSPI and SSFR), which include all ambient aerosol (as noted above, RSP models fine and coarse mode aerosol separately). The SSA values are also within the instrument uncertainty (4STAR) and variability (in situ)

ranges.

Figure 6 shows the AAOD and AOD for this same case. The two 80m cases have slightly higher AOD than the 1km measurements at 400nm (0.45 versus 0.42), but significantly higher AOD at 995nm (0.18 versus 0.11). Thus, the 80m values have markedly lower Ångström exponents—a difference of 0.4 for both SAE and EAE. This is again consistent with the aerosol between 80m and 1km (i.e. the boundary-layer aerosol) including a significant coarse-mode aerosol component – for

this region, very likely sea salt. Comparison with the 1km 4STAR sky scans is additionally instructive for this case as due to the very low absorption data in the boundary layer, the SSA values were not reported for these altitudes due to low signal-to-noise; thus the in situ values in Figure 5 are effectively averaged over only plume altitudes. Given that RSP and 4STAR both measure the full column we expect the retrieved column SSA to be somewhat higher than that from the measurements/retrievals of SSA for the aerosol for the BB plume altitudes only; this could be contributing to the higher SSA from RSP and to a lesser degree

from 4STAR as compared with in situ instrumentation (Figure 5). Based on this we might also expect the 4STAR retrieved SSA to be higher for the retrievals from 80m than from 1km, and this is not apparent except at wavelengths 870nm and longer. However, this could potentially be explained simply by a smaller relative contribution of boundary-layer aerosol to the total aerosol loading, particularly at the shorter wavelengths (i.e. the smaller difference between the two sets of sky scans at shorter wavelengths).

The AAE and EAE are given by the slopes of the AAOD and AOD versus wavelength, and the SAE can be inferred from the two (Figure 6). In comparing with the other instruments in this case study, the 1km altitude 4STAR EAEs and SAEs are more comparable to those derived from the in situ instruments. The 4STAR SAE and EAE values from 80m (including sea salt) are lower than the corresponding RSP Ångström exponents of $AOD_{fine+coarse}$. The 4STAR 1km values (plume only) agree with the RSP AOD from fine mode only for mid-visible values and diverge for wavelengths greater than 700nm, which may

somewhat be expected given the lower signal at longer wavelengths. While RSP observes slightly higher AOD (compared with





4STAR) at shorter wavelengths, at the longer wavelengths the RSP AOD is less than the 80m values (i.e. 4STAR measurements with additional coarse mode aerosol), but still greater than the 1km values. The in situ AOD proxy is notably lower, likely due to the inlet limitations for larger particles as discussed above.

In terms of AAE, the RSP results give higher values (1.25-1.26) than 4STAR, similar to the AAE from the PSAP (1.23). The average 4STAR $AAOD_{400nm}$ measured at 80m is 0.057 and is 0.049 at 1km, and $AAOD_{995nm}$ is 0.025 versus 0.023 at 80m and 1km, respectively, though the latter value is an average between 0.024 and 0.022. The resulting 4STAR AAE values for 80m versus 1km are more comparable than the pairs of SAE or EAE values: 0.79 vs 0.89, consistent with minimal contribution of sea salt to aerosol absorption. However, these values overall are notably lower than may be expected from theory; a more detailed discussion of the AAEs over the full campaign is found in Section 3.3 below.





**Figure 5.** Spectral SSA from 4STAR (blue squares), RSP (goldenrod triangles), and in situ (green diamonds) calculations for the case on 12 September 2016. This case was centered at approximately 18.0°S, 8.0°E, between 13:44 and 14:34 UTC. The bars on the in situ measurements represent the 10-90th percentile ranges for the profile considered. The two 4STAR curves with high $SSA_{995nm}$ are from the 80m altitude scans, while the curves with low $SSA_{995nm}$ curves are from the 1km altitude scans.





**Figure 6.** Spectral a) AAOD and b) AOD from 4STAR, RSP, and the in situ calculations for the case on 12 September 2016, as well as the AAE, SAE, and EAE values (derived from the slopes in log-log space of AAOD and AOD). RSP EAE and SAE include fine+coarse mode AOD, whereas RSP AAE are for fine mode only. Note that the in situ vertical profile for this case extended from 80m to 5.8km and may have undersampled the coarse-mode sea salt at lower altitudes due to inlet efficiency, whereas the two remote sensing instruments give full-column values above the P-3 (4STAR) to top-of-atmosphere, or below the ER-2 (RSP), extending to the surface. For the given 4STAR Ångström exponents, the first set refer to the sky scans from 1km, and the second refer to the scans from 80m, immediately before the aircraft ascent.



### 3.2.2 Case study: 20 September 2016 (cloudy scene)

The flight on September 20 (P-3 RF11) was focused on measuring atmospheric radiation with two parallel N-S flight lines along 9°E and 10.5°E. We again identify two cases which are suitable for the instrument comparison (one on each longitude line, 10.5°E and 9°E). Here we focus on the second of these two cases (P-3 RF11.2), which allowed for comparison of SSA

measured by the in-situ instruments and 4STAR on the P-3 with retrievals of SSA from both AirMSPI (on the ER-2) and SSFR (on the P-3) (as with RF08, both cases are included in the analysis in Section 3.3). This case was centered at approximately 16.7°S, 9°E and began at 10:45 UTC with a partial profile descent (ramp) from plume level (4.3km) to above-cloud (600m), followed by six sky scans at above-cloud/below-plume altitude between 10:59 and 11:08 UTC, with below-aircraft scene albedo between 0.45 to 0.62. Of these, two scans passed QC (the others were excluded due to high solar elevation on almucantar scans

and/or high error in the retrieval results). After the above-cloud leg, there was one full-profile square spiral maneuver from approximately 11:52 to 12:15 UTC which provided the observations used in the SSFR retrievals. Three ER-2 overpasses occurred along this longitude line; the AirMSPI retrievals are from overpasses during the first (northward; 11:41-11:51 UTC) and second (southward; 12:18-12:25 UTC) passes. The HSRL-2 lidar retrievals show that this case consisted of an aerosol layer of primarily smoke with contribution by dust of 10-15% of the 532 nm AOD. Measurements from the HiGEAR Aerodynamic

Particle Sizer (APS) also saw dust during this profile, and indicated that <10% of scattering was due to dust. For this case, the above-BL RH was somewhat greater and also more varied with altitude compared with the case from 12 September: between 10% and 80% through the atmospheric profile (water vapor mixing ratios ranging from 3500 to 13500 ppmv), though RH remained below 40% except for the plume maximum between 4.2km and 5.6km. Again we note that while the remote-sensing (4STAR, AirMSPI, and SSFR) SSAs are measured at ambient humidity, the in-situ (PSAP+Neph and PTI+Neph) values are

for aerosol dried to RH< 40%. While this difference was minimal for the first case (12 September) due to the generally lower RH, it has the potential to be higher for the second case (20 September). We discuss the implications of aerosol humidification in more detail in Section 4.2.

Figure 7 shows the spectral SSA from each instrument available for this case. We first note the general agreement between most instruments (within the stated uncertainties). While 4STAR and AirMSPI both show SSA decreasing at longer

wavelengths, 4STAR reports slightly lower SSA values (particularly at longer wavelengths) than AirMSPI. This is a common feature seen in most of the comparison cases (e.g. Figures 9a-c). The in-situ observations have less wavelength range than 4STAR and AirMSPI, but again show a decrease in SSA at its longer wavelength. The rate of decrease from 530nm to 660nm is seen to be greater than from 470nm to 530nm (this is also seen in the Ångström exponents calculated from 2-wavelength pairs, discussed in Section 3.3). As in Figure 5, the wavelength-specific Ångström exponents give a different SSA spectral

shape with a small maximum at 530nm (Figure A2), which was more pronounced here than in the case in Figure 5, possibly due to the greater aerosol loading in this case. The PTI+Neph $SSA_{530nm}$ is higher than the PSAP+Neph SSA, a result which is expected in light of the lower reported absorption by the PTI (Figure 4). Additionally, due to the limited PTI data at lower (below-plume) altitudes, the values going into the average value are likely biased towards the higher altitudes of the plume





itself. As PSAP+Neph SSA values were higher at higher altitudes (not shown), this sampling pattern may also contribute somewhat to the PTI's higher SSA relative to the column-average extinction-weighted PSAP+Neph.

The SSFR retrieval largely agrees with 4STAR, AirMSPI, and the in-situ data (again within the given instrument uncertainties) for mid-visible wavelengths, though it reports lower SSA for the shorter (<440nm) wavelengths. The spectral shape is
somewhat more spectrally flat but is consistent for all retrieved wavelengths to within the instrument uncertainty. Note that the retrieval at each individual wavelength is performed separately for SSFR. This is in contrast to 4STAR, AirMSPI, and RSP which involve assumptions about aerosol size distributions and particle shape. In addition, for these instruments the spectral refractive indices are retrieved by fitting all wavelengths simultaneously: a spectral smoothness constraint is imposed on the real and imaginary parts of aerosol refractive index to improve the retrievals. For 4STAR, this means that the inversion algorithm
may converge to a different result based on small perturbations in the wavelength-dependent AOD and radiance values used as inputs; this range is what we attempt to encapsulate within the stated uncertainty bars. For AirMSPI, the polarized radiances measured in the 470, 660, and 865 nm bands have less sensitivity to aerosol refractive index and coarse mode aerosol size than to aerosol loading and cloud microphysical properties. As a result, the AirMSPI retrieval accuracy of refractive index and coarse mode aerosol size distribution are subjected to greater measurement errors than the AOD and cloud microphysical part
of the retrieval, which potentially leads to errors with SSA (cf. the error bars with SSA). In contrast, the SSFR retrieval could be considered a somewhat more "direct" derivation of SSA, in that it retrieves SSA directly and individually for each wavelength on the basis of absorbed irradiance and 4STAR-measured AODs alone, with minimal constraints and assumptions on aerosol (e.g. size distribution, shape, and mixing state). At the same time, we note that the wavelength-dependence of SSA and asymmetry parameters which are retrieved from the SSFR measurements are not necessarily consistent with a physically realizable
microphysical model, which may well be a drawback in trying to generalize these retrievals for related radiative transfer calculations—though particularly in complex combined aerosol-cloud scenes such as these, the positives may outweigh the potential negative aspects of this retrieval method. Regardless, the fact that retrievals using such different measurement approaches agree so well is encouraging.

Figure 8 shows the spectral optical depths for this same case, by instrument. Note that the AOD input to the SSFR retrieval
($AOD_{400nm} = 0.78$) is obtained by 4STAR at the location of the square spiral, while the AOD reported as 4STAR (Figure 8b) are from the times of the sky scans on the southbound leg directly preceding the spiral, as described above. The first scan ($AOD_{400nm} = 1.05$) was approximately 50 km from the square spiral location, which accounts for the difference in AOD between the two. The second scan ($AOD_{400nm} = 1.24$) is approximately 110 km from the spiral location (8 minutes after the previous scan in flight time). While this is on the higher end of our desired spatial spread, we show both scans here to show
the good agreement in the spectral shape of the aerosol optical properties (i.e. spectral SSA and Ångström exponents, Figures 7 and 8) between the two retrievals even with differences in aerosol column loading (AOD and, consequently, AAOD) (Figure 8). Indeed, the Ångström exponents for this case vary within only 0.2 for all five instruments, which is better agreement than the three instruments shown in the previous case. The outlier again is the 4STAR AAE values of approximately 0.9, lower than any other retrieval. This is discussed in more detail in the following section.





**Figure 7.** As in Figure 5, but showing spectral SSA for the ORACLES instruments operating on 20 September 2016: 4STAR sky scan retrievals (blue squares); AirMSPI retrievals (pink triangles); SSFR square spiral (black circles); and profile-averaged extinction-weighted SSA from the PSAP+Neph (green diamonds) and PTI+Neph (purple star). This case was centered at approximately 16.7°S, 9°E, and sampled from 10:45 UTC to 12:15 UTC.

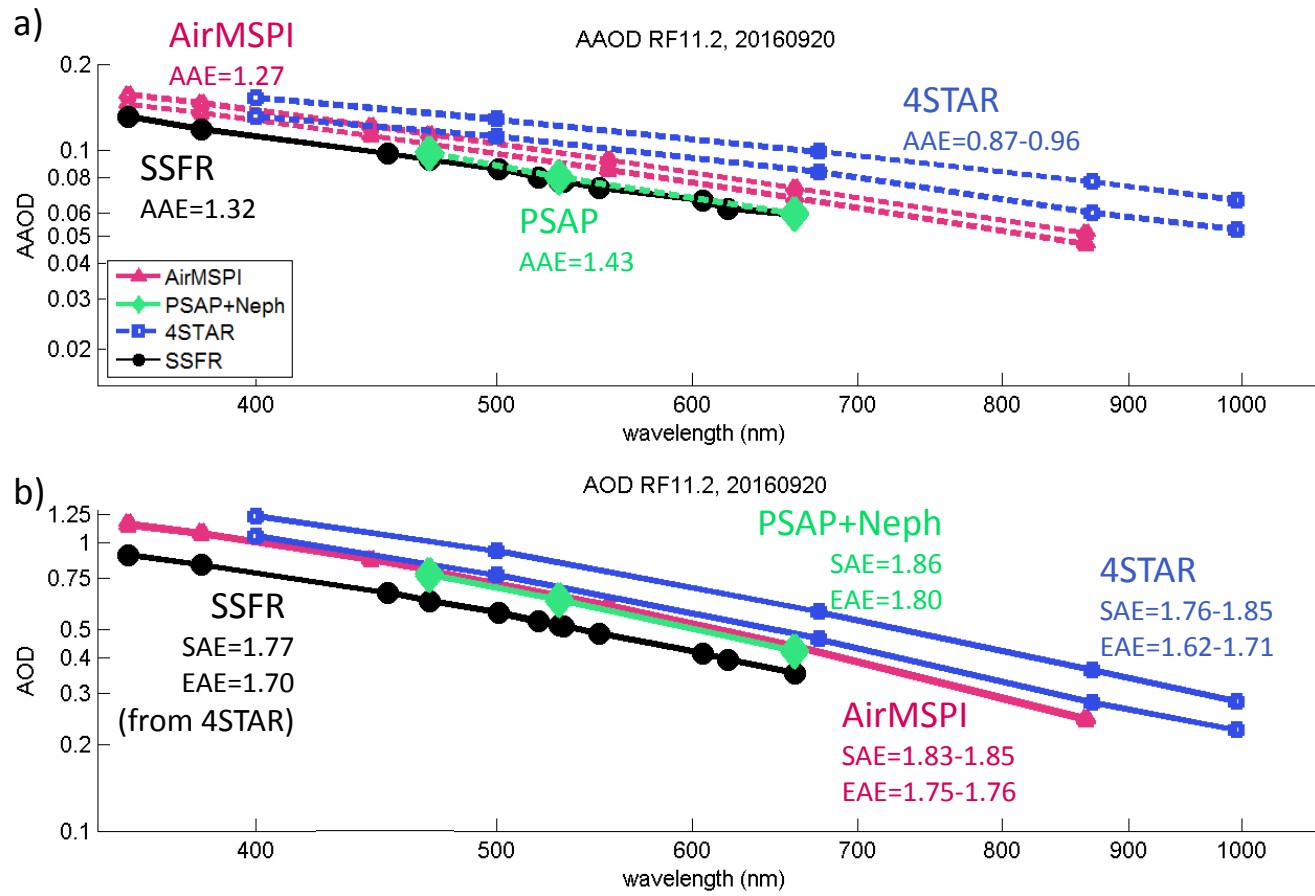

**Figure 8.** As in Figure 6, a) AAOD and b) AOD versus wavelength (both on log-log scales) for the case on 20 September for 4STAR, AirMSPI, SSFR, and PSAP+Neph. The PSAP+Neph optical depths are calculated from a 100m-bin average profile of nephelometer scattering and PSAP absorption, and integrated over the vertical column. This case had a full profile from 367m to 7.1km (shown) and additional partial profiles which appear in Figure 7. Note that SSFR uses 4STAR AOD as input; this coincident AOD is labeled as SSFR for clarity; thus, the differences between these three AOD spectra are due to spatial variability.



### 3.3 Campaign-wide ORACLES-2016 instrument comparisons

The two cases shown in the previous section are intended to give an idea of the range of observed cases in clear and cloudy skies, while allowing comparison with available RSP and SSFR retrievals. Next, we broaden scope to consider instrument comparisons for the campaign as a whole. We focus on in situ versus 4STAR comparisons, as these are the two methods with

enough coincident measurements to allow for a statistical examination. We also include a more limited number of comparisons using AirMSPI data where available.

Figure 9 shows scatter plots of several parameters (SSA, EAE, SAE, and AAE) as measured by sets of two instrument pairs—4STAR versus in situ versus AirMSPI—across the full set of comparison cases. As one case may have multiple retrievals (i.e. multiple vertical profiles, sky scans, and/or sweeps), the case-average data are indicated by filled black markers. All data

(i.e. multiple individual retrievals included in a comparison case) are shown as small x-marks, with grey lines indicating the range of retrievals within a given case. The top row of Figure 9 shows scatter plots of $SSA_{530nm}$ reported by 4STAR, in situ, and AirMSPI for the aggregation of all comparison cases where two of the three can be compared. 4STAR SSA is generally lower than the in situ SSA (Figure 9a), and AirMSPI SSA is higher than both 4STAR or in situ (Figure 9b,c). While 4STAR and in situ generally track one another, the relationship is not statistically significant (R=0.23, p=0.33). Similar results are seen

for EAE and SAE (Figure 9d-i), though the correlation in SAE between 4STAR and in situ is the only combination that could be considered robust: $R_{SAE}$=0.66 with p<0.01, whereas $R_{EAE}$=0.44 and p=0.06.

It is interesting to note that the one outlier in Figure 9d and g (low 4STAR SAE and EAE) is the case from 12 September described earlier (Figs 5 and 6), where the two 4STAR sky scans at lower altitude (i.e. immediately before the in situ profile) are those which result in the very low AE values which depress the average for that case. As seen in Figs 5 and 6, there is no

such divergence in $SSA_{530nm}$ or AAE for this case. We also note that the correlation described above between 4STAR and in situ EAE disappears when this outlier case is excluded.

In contrast to the other parameters, the 4STAR AAE shows, if anything, a negative relationship with in-situ-derived AAE (Figure 9j), though this relationship is again not statistically significant ($R_{AAE} = -0.37$, p=0.12). We additionally note that the 4STAR-reported AAE values are low in general, often <1, and are split into two populations, one in fairly good agreement

with in situ (R=0.50, p=0.08) and one substantially offset to lower 4STAR AAEs relative to in situ, with a similar but less robust correlation (R=0.59, p=0.22). As a whole, the full 4STAR AAE data set is negatively correlated with the in situ values. The relationships between AAE from AirMSPI and either of the other two instruments (Figure 9k,l) are also not statistically significant, due to the small range in AAE values retrieved from AirMSPI. The variability in AAE for aerosol within a given case (date and location) is also often quite large; this, combined with the lack of correlation between instruments, may simply

reflect high uncertainty in derived absorption Ångström exponents from all methods.

As the 4STAR retrievals frequently show AAE values less than 1, this bears some discussion. Notably, Bahadur et al. (2012) also derived very low values of AAE (as low as 0.55 for pure BC) using the same AERONET retrieval algorithm. While some previous observational studies have allowed for AAE values less than 1 (e.g., Bergstrom et al., 2007; Lack et al., 2008), it is generally suspected that this is the result of measurement artifacts or instrument uncertainties. While pure BC is typically




considered to have an AAE of 1, studies have shown this may vary based on particle size (e.g., Lack and Cappa, 2010; Wang et al., 2016), and may be either higher or lower than this value. However, brown carbon typically has AAE greater than that of BC, and the 4STAR AAE values are lower than expected from theory even for pure BC (e.g., Schnaiter et al., 2005; Gyawali et al., 2009; Lack and Cappa, 2010) let alone a mixture. Thus, values of AAE for a mixed aerosol (smoke) that are

less than 1 (for the wavelength range 440-675nm) are suspect in that they run counter to theory and to many observations of ambient-aerosol AAE, including in this region (Bergstrom et al., 2007). As such, these results should be treated with caution, considering the compounding uncertainties (i.e., as the slope of another derived property) inherent in the derivation of AAE by any method.

Figure 10 shows the difference in Ångström exponents (AAE, top row; SAE, bottom row) from the in-situ measurements

versus from 4STAR retrievals as a function of AAE (top row) and SAE (bottom row) for each instrument, and as a function of AAOD (for AAE) or AOD (for SAE). This allows for visualization of the level of agreement between the two methods as a function of the retrieved properties themselves, and amount of aerosol loading.

The first two panels (Figure 10a,b) show the dependence of the difference between AAE from in situ and 4STAR measurements as a function of AAE. This shows a fairly strong correlation between higher values of in situ AAE and greater differences

(in situ minus 4STAR), with the expected opposite correlation observed for high 4STAR AAE versus AAE difference (R=0.76 and -0.89, respectively, both p<0.001). In other words, as AAE increases as measured by the in-situ observations, the difference between the in situ and 4STAR AAEs increases – and higher in situ AAE values correspond with lower 4STAR AAE values, resulting in a greater difference between the two (Figure 9j). The fact that the difference between the two is correlated with each suggests that this is not a clear case of one versus the other driving the large differences.

A weaker correlation of the same sign is observed for the difference in SAE and in situ- or 4STAR-measured SAE (R=0.48 and -0.57, respectively, both p<0.001, though the latter shrinks to R=-0.22, p=0.04 when removing the outlier shown) (Figure 10d,e). While there is a weak (R=0.50) but significant correlation (p<0.001) between AAOD and the AAE difference between the two instruments (Figure 10c), there appears to be no correlation between total aerosol loading (4STAR AOD) and SAE (Figure 10f). This suggests that total aerosol loading does not affect the instrument agreement, but higher values of absorbing

aerosol may bias one or both of the instruments; while the filter-based PSAP instruments have well-known artifacts, there is also a weak negative correlation between $AAOD_{530nm}$ and $AAE_{4STAR}$ (R=-0.34, p=0.001).

Figure 10 additionally shows the wavelength-dependence in AE calculated for different wavelength ranges (colored markers). This illustrates the difference in AE values (AAE and SAE), particularly from the in situ instruments, which are may result simply from altering the wavelengths used in their calculations, without significantly altering the wavelength limits.

Bergstrom et al. (2007) showed a similar dependence of the calculated AAEs on the wavelengths used. For example, here we see a distinct spread between the AAE calculated between the two shortest wavelengths (470 to 530nm; blue circles) and the two longest wavelengths (530 to 660nm; red triangles), with the former showing the largest difference between in situ and 4STAR as well as the largest values of in situ AAEs. The same is true for the in situ SAE, except with opposite sign: the SAEs calculated using the longest wavelengths have the highest values. The differences in AAE are largely positive regardless of

the wavelengths used: in other words, in situ gives higher AAE than 4STAR, with the 4STAR AAE values anomalously low





**Figure 9.** *(top row)* SSA$_{530nm}$, *(second row)* EAE, *(third row)* SAE, and *(bottom row)* AAE for *(left column)* 4STAR versus in situ measurements, *(middle column)* AirMSPI versus in situ measurements, and *(right column)* AirMSPI versus 4STAR measurements, for all comparison cases with valid coincident observations. All individual retrievals are shown by grey x-marks; for cases with multiple retrievals from a single instrument, the average value and range for a given case is shown by a black circle with whiskers.





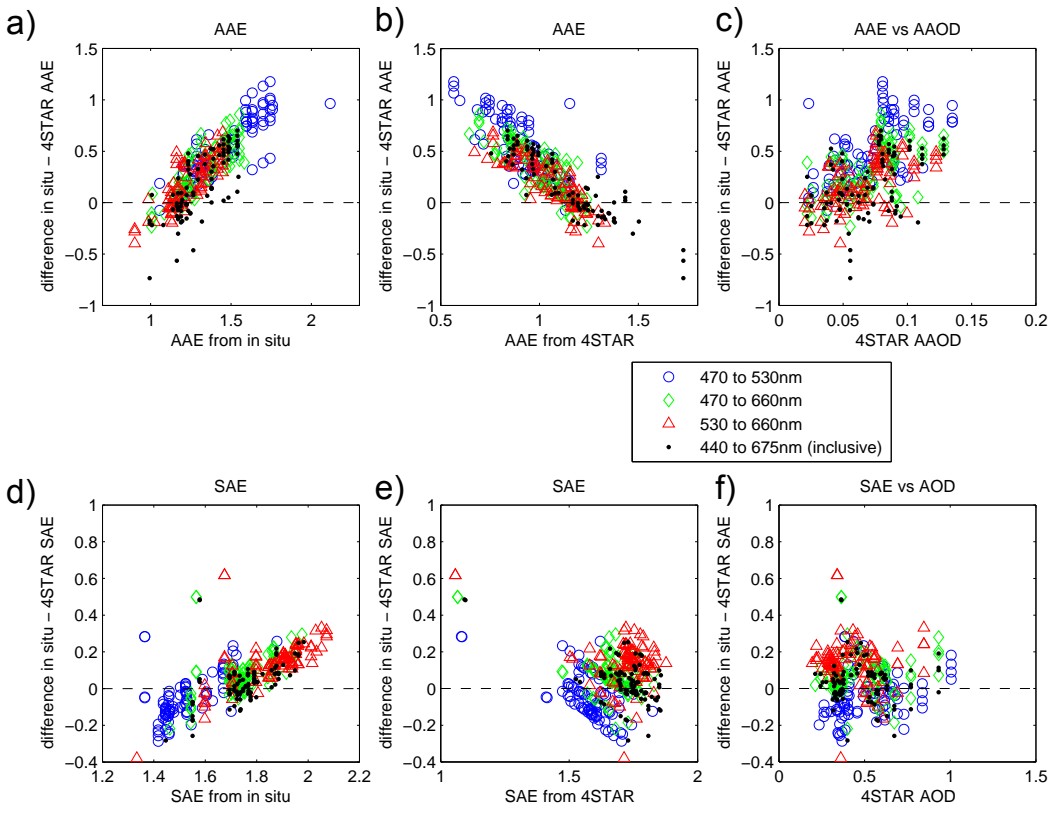

**Figure 10.** Difference between AAE measured in-situ versus retrieved from 4STAR (in situ - 4STAR) as a function of a) in situ AAE, b) 4STAR AAE, and c) 4STAR AAOD. The bottom panel shows the same, but for SAE differences versus d) in situ SAE, e) 4STAR SAE, and f) 4STAR AOD. Note that multiple individual points for one instrument (profiles or sky scans) may be shown for comparison cases which included >1 sample per instrument (i.e. x-marks in Figure 9). This is an indication of the variability within a specific case.

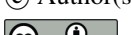



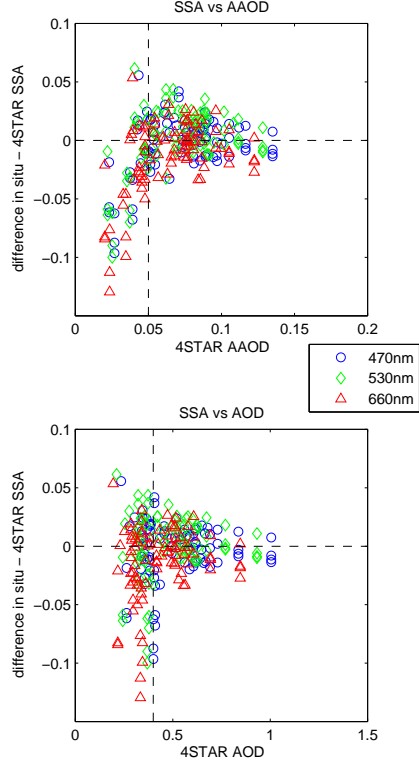

**Figure 11.** The difference between SSA measured in-situ versus retrieved from 4STAR (in situ - 4STAR) as a function of 4STAR *(top)* AAOD and *(bottom)* AOD, respectively.

relative to previous estimates of AAE, as discussed earlier. For SAE (and also EAE), the differences are more symmetrical, with negative values (4STAR greater than in situ) for the shortest wavelengths, and positive values (in situ greater than 4STAR) for the longest wavelengths. It is also worth noting that the difference between values for these two wavelength ranges (shorter minus longer) is between 0.1 and 0.4 for AAE and up to -0.5 for SAE from in situ, and up to -0.3 for AAE and SAE from

5   4STAR, which is a substantial range, given the range in instrument AEs over all the comparison cases were largely within 0.5 of one another. The significant variability seen here between values simply calculated from different wavelengths suggests caution not to overinterpret Ångström exponents, particularly those calculated using only two wavelengths.

    In contrast to the AE results, there is no dependence in the differences in mid-visible SSAs between the in-situ measurements and 4STAR retrievals: for $AOD_{470nm} > 0.4$ and $AAOD_{470nm} > 0.05$ (Figure 11) the differences are within $\pm 0.03$,

10  within the expected uncertainties. At lower AOD/AAOD the differences are more pronounced, tending to higher values for 4STAR at lower loadings. This is consistent with the minimum AOD threshold value defined in the AERONET QC procedure ($AOD_{440nm} > 0.4$).



## 4 Discussion

### 4.1 Campaign-wide measurements of SSA from multiple instruments

We now turn to the full set of SSA data from ORACLES-2016. Figure 12a shows campaign-wide averages of SSA for 4STAR, AirMSPI, and PSAP+Nephelometer. As these are campaign-wide values, they are not strictly comparable to one another; for

example, the spatial coverage of the AirMSPI retrievals is larger than the other two instruments, due to the greater spatial range of the ER-2 versus P-3 flights (Figure 12a, inset). Also, the AirMSPI data considered here are from 6 flight days, compared with 13 flight days for 4STAR and 14 days for the in situ measurements. In addition, the P-3-based measurements (PSAP+Neph and 4STAR) were able to sample more coastal aerosol, which may be more influenced by variability in local aerosol sources and thus composition (compared with far-from-coast flights which would sample the upper-level plume of more uniform origin).

Further discussion of the 4STAR-observed temporal and spatial variability of aerosol loading and size in ORACLES-2016 may be found in LeBlanc et al. (2019), and comprehensive discussion of the observations as they compare with model outputs will be included in a later work. Interestingly, both the offset and the sign of the difference between 4STAR and in situ measurements are similar to the results of Schafer et al. (2014), who found AERONET-derived SSAs were on average 0.011 lower than in situ (PSAP+Nephelometer) values for flights over the eastern US.

Figure 12a also shows that the variability of the SSA from the P-3-based measurements (4STAR and in situ) is fairly similar in the mid-visible range: roughly 0.025 between the first and third quartiles for 4STAR versus 0.028 for the in situ measurements. At longer wavelengths (875 and 995 nm), the 4STAR variability approximately doubles; the uncertainty for individual 4STAR sky scans is also higher at these longer wavelengths. In-situ data are not available at these wavelengths.

    Figure 12b shows the ORACLES spectral SSA values compared with those from previous studies, as presented in Figure

1. SSA from the three ORACLES measurements are within the range of previous observations of SSA, with the difference in SSA between the SAFARI "fresh" versus "aged" plume (Haywood et al., 2003) bounding these observations. It is important to note that this "fresh" plume is based on a single flight directly over a terrestrial emission source (13 September 2000), whereas the "aged" values are the mean values from the remaining 8 flights, ostensibly sampling aerosol at least 2-3 days old (Haywood et al., 2003, their Table 2). While the "aged" SAFARI values should be more comparable to the expected age of ORACLES-

sampled aerosol from oceanic overflights (compared with the "fresh" plume), in-field experience with aerosol model forecasts indicated the age of ORACLES-sampled free-tropospheric aerosols were typically older than four days (Dobracki et al., 2019). Both of these ORACLES and SAFARI values are based on in situ measurements (PSAP+Neph), whereas the SAFARI result shown in Russell et al. (2010) (and also described in Bergstrom et al., 2003) is from an SSFR-centered retrieval combined with data from a precursor to 4STAR, AATS-14 (the Ames Airborne Tracking Sunphotometer at 14 wavelengths) similar to

the retrieval used in this paper. The retrieval is from a radiation wall within a single flight (6 September 2000). By way of comparison, flight-average SSA from the PSAP+Neph measurements on that same flight was given as 0.87, 0.86, and 0.83 at 450, 550, and 700 nm (Haywood et al., 2003), somewhat lower than the "aged" average of 0.91, 0.90, and 0.87 and only 0.01 lower than the SSFR values for the same flight. In the same work, SSA was also derived using filter measurements integrated over PCASP size distributions; the PCASP-derived values for this particular flight are given as 0.92, 0.89, and 0.87, closer to





the high end of the range. Thus, even with this small amount of data, it is difficult to definitively say whether the variability among previous measurement methods is purely a function of natural variability, or whether it is predominantly due to systemic variability between different measurement methods.





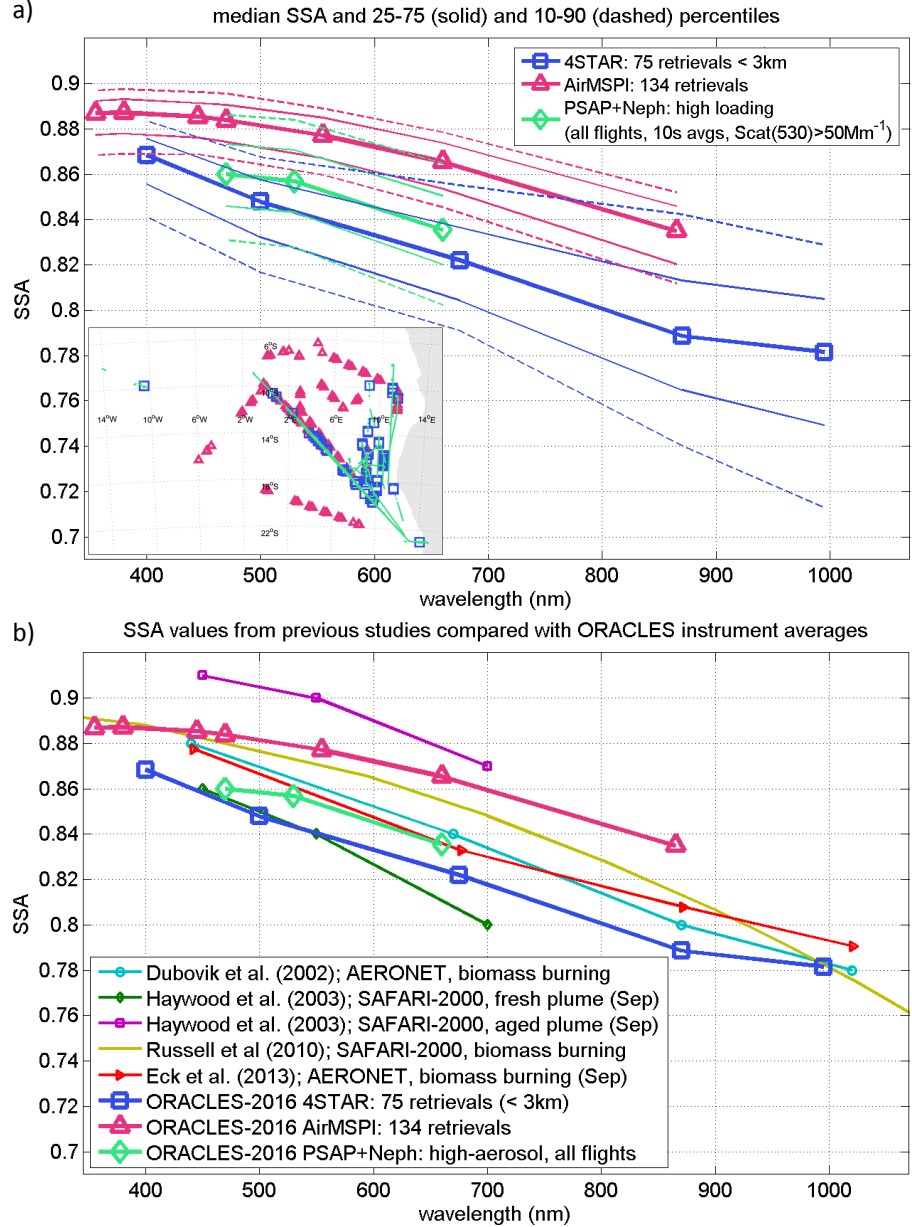

**Figure 12.** a) SSA from all measurements/retrievals over ORACLES-2016, from 4STAR, AirMSPI, and the nephelometer and PSAP in-situ data. The campaign-wide median is indicated by the thick solid line, 25-75th percentiles are indicated by thin solid lines, and 10-90 percentiles are indicated by the dashed lines. Note that the medians of the different instruments are not strictly comparable, as the campaign-wide averages are calculated using more data than just the coincident cases. The inset shows the spatial distribution (lat/lon) of measurements by the three instruments. b) The median SSA by instrument as shown in a) compared with SSA from previous studies in this same region (Figure 1).



## 4.2 Impacts of humidification on aerosols

In this work we have presented SSA as calculated from in situ measurements of scattering (from a nephelometer) and absorption (from a PSAP and a PTI); all are for aerosol dried to RH$< 40\%$. It is well-known that an increase in RH will result in an increase in scattering, which, if it is the sole consequence of the humidification, will act to increase the SSA. However, humidification

of aerosol will impact both scattering and absorption, likely not to the same degree, and thus will affect the resulting SSA, potentially in competing directions. For this reason, we have presented in situ measurements as measured (i.e. dried) in this present work.

Aerosol absorption is likely affected by humidification, but the magnitude of this effect is much less well understood, due in large part to the difficulty in measuring humidified absorption directly (e.g., Arnott et al., 2003). However, several studies have

attempted to address this question. One theory-based study using a BC core/shell model (Redemann et al., 2001) found that in terms of the aerosol absorption, the absorption enhancement at RH=80% was approximately a factor of 1.1, resulting in a decrease in SSA of 0.02 at 550nm. They also found that the degree to which humidification enhanced absorption is dependent on the aerosol size distribution, with absorption enhancement as high as a factor of 1.75 for certain size and humidity conditions (up to 99.5%). In a later lab-based study measuring total extinction and nephelometer-measured scattering, Brem et al. (2012)

found enhancement of absorption from biomass-based OC aerosol to be significant at 467 and 530nm for RH$> 85\%$ (the 660nm data were within the instrument noise). Between 32% and 95% RH, the absorption at 467 and 530nm increased by factors of 2.2±0.7 and 2.7±1.2, greater than that found by Redemann et al. (2001); combined with the observed enhancement in scattering, this corresponded to a change in SSA on the order of 0.06 for 470nm and 0.03 for 530nm, though the authors acknowledge that their method is subject to large uncertainties. We further note that the majority of studies considering the

effect of humidification on aerosol absorption (and scattering) consider the more extreme values of RH$\geq 85\%$ or higher, which was a rare occurrence in the BB plume sampled in ORACLES-2016 (observed plume RH was a maximum of 80% and often lower, as described above). The magnitude of these effects are thus likely to be some factor smaller than the values reported in the literature. The unequal enhancement factors at different wavelengths suggest that the absorption Ångström exponents would also be affected. As the effects of humidification on absorption have been seen to depend on aerosol age and composition

(i.e., hydrophobic vs hydrophilic; coated vs uncoated aerosol) (e.g., Mikhailov et al., 2006; Zhang et al., 2008), and even result in a decrease in absorption with increasing humidity (by 20% for an RH of 80%, in Lewis et al., 2009) the question of the magnitude of RH impacts on absorption remains an open one.

Due to this uncertainty in the cumulative effects of humidification, an open question is how humidification affects the study at hand, given that we perform comparisons between in-situ SSA values from measurements at low RH and the SSA

values from the retrievals (SSFR, 4STAR, AirMSPI, RSP) which are made at ambient RH. While there was persistent elevated relative humidity within the biomass burning plume relative to the free troposphere absent the plume, in ORACLES-2016 approximately half of the time where aerosol concentrations were high ($\sigma_{\mathrm{scat,530nm}} >$50Mm$^{-1}$) – i.e. when the P-3 was in the plume – the measured relative humidity was $< 40\%$ (the "dry" threshold), and an additional 30% of the time RH was in the range 40%<RH<60%. While we cannot make a reasonable estimate of the effects of humidification on absorption, we





can estimate how much it might be affecting scattering, and bound the upper limit on SSA accordingly (i.e. assuming zero effect on absorption). Based on a campaign-average scattering enhancement of 1.4 for a plume RH of 80%, and making the coarse assumption of no effect of humidity on absorption, we would expect the maximum impact of typical humidification on scattering to increase instantaneous SSA (at 530nm) by a maximum of 0.03-0.05. However, in reality, this value will be lower,

first due to the variability of RH with altitude as described in Section 3.2 which will have a lesser impact at lower altitudes and thus on the column-averaged values considered here; and second due to the competing impacts of humidification on absorption as described above, which likely have the opposite effect of lowering SSA. Due to the high uncertainties surrounding these competing effects, we leave it to a future work to provide more a detailed quantification of humidification effects on the SEA biomass burning SSA.

## 5   Conclusions

In this work we present new measurements of absorbing aerosol optical properties over the southeast Atlantic Ocean, a region with a significant and persistent seasonal biomass burning plume overlying stratocumulus clouds and which, up to now, has had a dearth of observations. For specific comparison cases, the retrievals from remote sensing (4STAR, AirMSPI, RSP, and SSFR) and in situ (derived from PSAP, Nephelometer, and PTI) agree within given uncertainty ranges, though with some indications

of systematic differences between the different methods. Specifically, the modified AERONET retrievals applied to 4STAR data typically produce the lowest SSA, while the AirMSPI polarized retrieval generally yields the highest SSA. Correlations between individual instruments over an aggregate of cases (using between 9 and 19 available comparison cases for different instrument pairs) were not significant in most cases, with the exception of a weak positive correlation between 4STAR- and in situ-derived SAE). AAE is the least certain of the retrieved absorption properties, as it shows a weak, yet negative correlation

between the two instruments considered, with a significant portion of the data reporting AAE values less than 1.

In terms of the ORACLES-2016 dataset as a whole, we find median SSAs from 4STAR to be 0.87, 0.85, 0.82, 0.79, and 0.78 at 400, 500, 675, 870, and 995nm; from AirMSPI to be 0.88, 0.87, and 0.84 at the retrieved wavelengths of 470, 660, and 865nm, and from in situ measurements to be 0.86, 0.86, and 0.84 at 470, 530, and 660nm. Campaign-wide data variability (5th-95th percentiles) are roughly equivalent for 4STAR and in situ measurements in the mid-visible ($\pm$0.03), and is greater

at the longer 4STAR wavelengths ($\pm$0.05 at 870nm). The AirMSPI data exhibit less variability at all wavelengths ($\pm$0.015 at 470nm and $\pm$0.02 at 865nm). While these are not directly comparable to one another due to differences in spatial and temporal sampling of the different instruments, they give an indication of the best estimate and range of biomass burning SSAs over the southeast Atlantic Ocean. The range of SSA values reported between different instruments during ORACLES is consistent with the range reported among previous observational studies over this region, but slightly higher than those reported on Ascension

Island (Zuidema et al., 2018). In Section 1.2, we discussed the radiative forcing implications of SSA; finally, we note that the range of SSAs observed from each instrument is of the magnitude expected to change local direct radiative effects by 10-20W/m$^2$ (SSA$\pm$0.03), which may give an indication of the impact of the results. Any studies which rely on a prescribed set of aerosol properties — such as SSA — as input, should thus consider a realistic spatiotemporal distribution of aerosol optical





properties in order to best capture the reality of the aerosol conditions over this region. It is important to take into account the impacts of the spatial variability and uncertainty from a given instrument, as they may affect resulting determinations of radiative effects of biomass burning aerosols.

*Author contributions.* JR and PZ are PIs for the ORACLES campaign, with SD on the deployment leadership team and providing advisory
expertise. SB and RF provided HSRL-2 data. BC, XL, SS, and BvD provided RSP data. SC and KSS provided SSFR data. KP, CF, MK, SL, and MSR provided 4STAR data. SF and SH provided HiGEAR data. AS provided PTI data. GVH and FX provided AirMSPI data. YS was ORACLES data manager and created campaign merge datasets. KP performed the bulk of the comparative analysis and wrote the paper, both with input from the other authors.

*Competing interests.* The authors declare no competing interests.

*Acknowledgements.* ORACLES is funded by NASA Earth Venture Suborbital-2 grant NNH13ZDA001N-EVS2. We thank the ORACLES deployment support teams and the science team for a successful and productive mission.

*Code and data availability.* The data used in this paper are publicly available at http://dx.doi.org/10.5067/Suborbital/ORACLES/P3/2016_V1 for the P-3 data (4STAR, in situ, SSFR) and at http://dx.doi.org/10.5067/Suborbital/ORACLES/ER2/2016_V1 for ER-2 data (RSP). All AirMSPI L1 data are available at http://doi.org/10.5067/AIRCRAFT/ORACLES/RADIANCE/AirMSPI, and the AirMSPI above-cloud
aerosol (ACA) retrieval data used here may be found at https://eosweb.larc.nasa.gov/project/airmspi/preliminary-airmspi-datasetsunderXu_etal_JGR2018_ACA_AirMSPI. The codes used in processing 4STAR sky scan data may be found at https://doi.org/10.5281/zenodo.1492912.



## Appendix A: PSAP Virkkula corrections

As discussed in the above text, the PSAP absorption corrections were applied as given in Virkkula (2010). Corrections of this nature must be applied due to the limitations inherent in filter-based measurements. Virkkula (2010) gives both wavelength-specific (470, 530, and 660nm) corrections, and a wavelength-averaged correction (his Table 1). We choose the latter due to the

5  discovery during the LASIC campaign (Zuidema et al., 2018) that using wavelength-specific values resulted in an unphysical jump in AAE upon changing of the filters. The difference between the absorption coefficients, and the resulting SSA, for the two methods is shown in Figure A1. Note that the wavelength-specific values generally show $\mathrm{SSA}_{470} < \mathrm{SSA}_{530}$ due to higher reported absorption at 470nm. Figure A.2 highlights the difference in wavelength-averaged versus wavelength-specific SSA for the individual profiles in the 20 September case, as discussed in the text. Note that for the analysis in Section 3, the Virkkula

10  corrections likely do not contribute to the observed differences between PTI and PSAP, as the SSA at the 530nm wavelength is less impacted by the choice of correction factor.



**Figure A1.** Normalized nephelometer scattering (left), PSAP absorption (center), and resulting SSA (right) for 470, 530, and 660nm, using wavelength-specific (top) and wavelength-averaged (bottom) Virkkula corrections. Values are normalized to highlight the relative spectral shape. Blue squares indicated the median of all observations (black). Note that while the scattering and absorption both exhibit an overall decrease with wavelength, the much sharper decrease in absorption results in a small maximum in SSA at 530nm under the standard wavelength-dependent Virkkula corrections applied to the PSAP. This spectral feature disappears in the average Virkkula case.




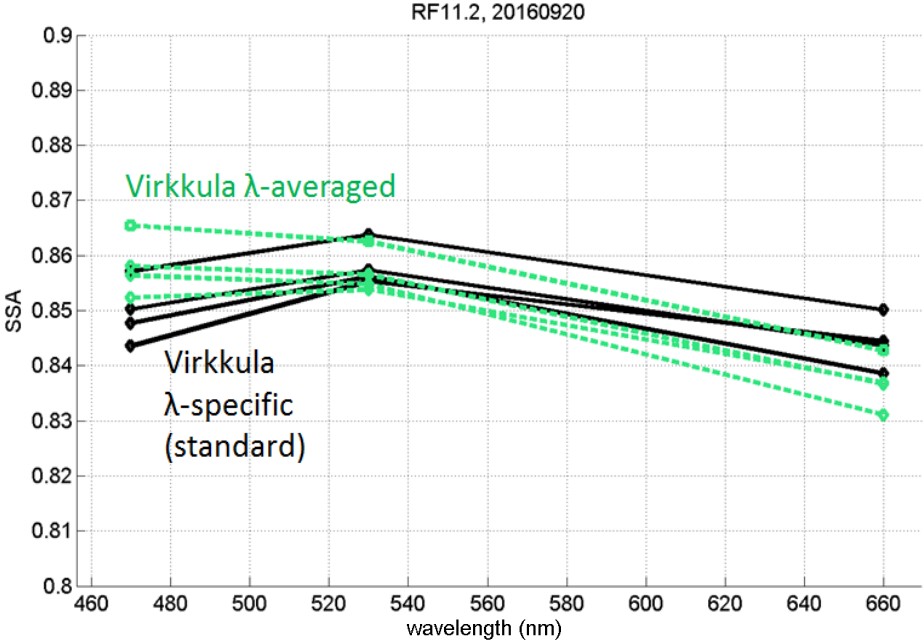

**Figure A2.** Profile-averaged extinction-weighted SSA for the case study shown on 20 September 2016. The impact on the 470nm SSA (and consequently, the spectral shape) is the most notable difference. We attribute this spectral shape to the stronger increase in aerosol absorption at the shortest wavelength– while scattering also increases, it exhibits a smaller wavelength dependence than does absorption (Figure A1).

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
