# Peer review of "Intercomparison of biomass burning aerosol optical properties from in-situ and remote-sensing instruments in ORACLES-2016"

_Atmospheric Chemistry and Physics, 2019_

## Referee Comment (RC1) · Anonymous Referee #1 · 21 Mar 2019

This manuscript presents the comparison of biomass burning aerosol optical properties namely single scattering albedo and Ångström exponents measured from in-situ and remote sensing instruments during ORACLES-2016 campaign. They presented two different case studies and found that SSA agrees within measurement uncertainties between multiple instruments. Furthermore, they found that instruments agreement in more robust if AOD400 > 0.4. Even though two case studies show agreement between instruments, campaign-wide average and range show values are highly variable between instruments.

Overall, this manuscript is well written. This work is an important contributor to connect

in-situ and remote sensing measurement of biomass burning aerosol optical properties. I have some comments for improving the manuscript and that should be considered before the publication.

Major Comments:

Even in 30s average data, PSAP shows variation in absorption by the factor of almost 6 while PTI shows about only 2. Even though, authors conclude this is due to the higher noise level in PTI but still they used PTI data to compare with remote sensing measurements. I don't get the whole point of doing that. Authors keep talking about the improvement in the PTI measurements for 2018 deployment given improved PTI data is not included in this study. I would recommend either using the data from 2018 PTI deployment or don't repeat the statement. Increase in SSA at longer wavelengths in two 80m scan is concluded due to sea salt in the boundary layer. This fact is not evident in the AAOD plot (Figure 6a). AAOD for 4STAR (80 m) scan is larger than 1 km scan indicating there were absorbing aerosol on boundary layer too. Why RSP is different than 4STAR 80 m? As expected, SSA from in situ may not exactly be correlated with remote sensing given an impact of RH, why two remote sensing method does not correlate at all (Figure 9c)? Authors should explore that in more depth than just present the result. I am not much worried about the inconsistency between in situ vs remote sensing but why AirMSPI vs 4STAR does not show correlation with any of the optical properties discussed in this study? For example, AAE measured by 4STAR shows variations between 0.8 to 1.8 while AirMSPI AAE lies between 1.2 to 1.4. Since their conclusion for this variation is higher uncertainty in derived AAE, it is better to do in-depth uncertainty analysis than just to mention. Given the fact that tow remote sensing method has more inconsistency than the remote sensing and in situ, I recommend the author to explore similar relationship (as shown in figure 10) between remote sensing methods. What is the whole point of choosing remote sensing and in situ? Given in situ have unclear bias from the RH as discuss later by the authors. In addition, the difference should be mention in relative not in absolute.

Minor Comments:

Page 2 Line 12: Mixing state is also an important factor.

Page 2 Line 18: Absorbing aerosol also shown to impact the global circulation patterns (Brown et al., 2018) in addition to the facts included in line 16-20.

Page 2 Line 23: SSA is defined already.

Page 2 Line 24: "Typically between 0.7 and 0.95" this statement is not always true for BB smoke. There are lots of laboratory and ambient studies which shows a huge range of SSA from BB emissions (Liu et al., 2014; Pokhrel et al., 2016; Vakkari et al., 2014).

Page 11 Line 29: Is there a specific reason to calculate SSA by extinction weighted? What is the benefit of it over SSA calculated from vertically integrated absorption and scattering coefficients?

Page 12 Line 10: What was the noise level of 30s averages PTI?

Page 17 Figure 4: What is the slope of the fitting line?

Page 18 Line 23-24: Since bars on the figure represent 10-90th percentile ranges. Which factor (instrument uncertainty or variations in data) contribute more at a longer wavelength.

Page 19 Line 33: 4STAR 1km plume only or sky scan?

Page 21 Figure 5: Please use the same legend in Figure 5 & 6.

Page 23 Line 24-26: More explanation for possible reasons.

Page 23 Line 31: How aerosol loading impact the spectral dependence of SSA?

Page 23 Line 32: Do PTI and PSAP SSA's comparable at higher altitude cases only?

Page 24 Line 3: Why AirMSPI uncertainty increase by a huge amount from about 450 to 470 nm?

Page 25 Figure 7: Be consistent between graph and legend.

Page 31 Line 9: Please express difference in relative.

References Brown, H., Liu, X., Feng, Y., Jiang, Y., Wu, M., Lu, Z., Wu, C., Murphy, S. and Pokhrel, R.: Radiative Effect and Climate Impacts of Brown Carbon with the Community Atmosphere Model (CAM5), Atmos. Chem. Phys. Discuss., 18, 17745–17768, doi:10.5194/acp-18-17745-2018, 2018.

Liu, S., Aiken, A. C., Arata, C., Dubey, M. K., Stockwell, C. E., Yokelson, R. J., Stone, E. a, Jayarathne, T., Robinson, A. L., Demott, P. J. and Kreidenweis, S. M.: Aerosol single scattering albedo dependence on biomass combustion efficiency: Laboratory and field studies, Geophys. Res. Lett., 41, 742–748, doi:10.1002/2013GL058392, 2014.

Pokhrel, R. P., Wagner, N. L., Langridge, J. M., Lack, D. A., Jayarathne, T., Stone, E. A., Stockwell, C. E., Yokelson, R. J. and Murphy, S. M.: Parameterization of single-scattering albedo (SSA) and absorption Ångström exponent (AAE) with EC/OC for aerosol emissions from biomass burning, Atmos. Chem. Phys., 16(15), 9549–9561, doi:10.5194/acp-16-9549-2016, 2016.

Vakkari, V., Kerminen, V.-M., Beukes, J. P., Titta, P., Zyl, P. G. van, Josipovic, M., Wnter, A. D., Jaars, K., Worsnop, D. R., Kulmala, M. and Laakso, L.: Rapid change in biomass burning aerosols by atmospheric oxidation, Geophys. Res. Lett., 2644–2651, doi:10.1002/2014GL059396, 2014.

---

## Referee Comment (RC2) · Anonymous Referee #2 · 25 Mar 2019

The paper presents results from several instruments to measure or retrieve aerosol optical properties, especially aerosol absorption, single scattering albedo, and their wavelength dependency during the NASA ORACLES campaign in September 2016. The paper is well written, it is definitely worth publishing in ACP. Actually, I did not find any errors but some points I don't understand, see below. I also have some suggestions for improving the paper.

I tried to find basic info of the flights:

- Approximate flight altitude range of the two aircraft

- Thickness of plumes, flight time in plume

[Figure]

- Estimated age of the aerosol at the measurement location since emission from the fires

- Distance from coast and fires

- Range of scattering and absorption coefficients, RH, and AOD in the plumes

The info is within the text but very scattered. Consider giving these in a table where you present descriptive statistics of these properties. And combined to your Fig. 3.

There is the section " 4.2 Impacts of humidification on aerosols". The discussion concentrates on SSA, that discussion is ok. However, you used the in situ instruments also for calculating AOD. It is definitely underestimated by using dry in situ data. Actually, when I look at Figs. 6 and 8 it also looks like that. Discuss this. And I expected to see some figures on data collected with the RH-nephelometer. Why don't you use that? Some figure? How does the scattering enhancement vary in the plumes? Effects on AOD?

Detailed comments

P10L24, "... operational modes: step-and-stare view mode and sweep mode, " explain with a couple of sentences, not every reader knows these

P11L22-23 "SSA was calculated using the measured PSAP absorption combined with dried (RH<40%) TSI nephelometer scattering interpolated to PSAP wavelengths." Shouldn't you use ambient RH scattering when comparing with the remote-sensing instruments? The same applies to calculating AOD from the in situ data.

P12L12 "... and absorption (AAE) are calculated from PTI absorption for data within ... " How can you calculate AAE from a one-wavelength instrument? I must be missing something.

P13L19 " ... AOD, and absorption is then derived as the change in net irradiance over the AOD differential ... " Don't understand. Isn't the change of net irradiance due to

both scattering and absorption? If I am wrong, please add a short explanation.

Fig 3. Name the aircraft in the Fig. Draw approximate vertical and horizontal scales in km: plume and flight altitudes, plume thickness.

P17L9 "... The generally lower PTI absorption could be due to several factors..." How do you know which instrument to trust?

In section "3.2.1 Case study: 12 September 2016" the in-situ-derived AOD is clearly lower than that derived from the remote sensing methods (Fig. 6). It is explained by the inlet cutoff on P19. Then in section "3.2.2 Case study: 20 September 2016" the AODs are essentially the same (Fig. 8). Why is that? Why couldn't the explanation for 9/12 be an aerosol layer above the plume measured with the in situ instruments? Have you considered comparing differential AODs of the in situ and remote-sensing methods? That would help in eliminating possible error due to high-level aerosol not observed with the in situ instruments.

P27L28 "..due to the small range in AAE values retrieved from AirMSPI" Any explanations or hypotheses?

P28L4-5 " ... values of AAE for a mixed aerosol (smoke) that are less than 1 (for the wavelength range 440-675nm) are suspect in that they run counter to theory ... " This is not true, it is easy to show with a Mie code that AAE can be < 1 both for uncoated and coated BC particles. Have a look at Figs. 7-9 of Gyawali et al. (2009) and Fig 8 of Lack and Cappa (2010) and the related discussions in these papers.

---

## Author Comment (AC1) · 8 Jun 2019

We thank both reviewers for their thoughtful and constructive comments. In response to these, we have made a number of edits which we believe improve our manuscript. Attached, please find the reviewer's comments with our responses in bold. A marked-up revision of the manuscript is additionally attached, showing all changes (additions and subtractions) in response to reviewers, some minor grammatical edits, and a correction to the Figure 2 caption which was erroneously printed as the Table 2 caption. Unless otherwise specified, the line numbers cited below refer to this marked-up revised manuscript.

**Anonymous Referee #1**

This manuscript presents the comparison of biomass burning aerosol optical properties namely single scattering albedo and Ångström exponents measured from in-situ and remote sensing instruments during ORACLES-2016 campaign. They presented two different case studies and found that SSA agrees within measurement uncertainties between multiple instruments. Furthermore, they found that instruments agreement in more robust if AOD400 > 0.4. Even though two case studies show agreement between instruments, campaign-wide average and range show values are highly variable between instruments.

Overall, this manuscript is well written. This work is an important contributor to connect in-situ and remote sensing measurement of biomass burning aerosol optical properties.

I have some comments for improving the manuscript and that should be considered before the publication.

Major Comments:

Even in 30s average data, PSAP shows variation in absorption by the factor of almost 6 while PTI shows about only 2. Even though, authors conclude this is due to the higher noise level in PTI but still they used PTI data to compare with remote sensing measurements. I don't get the whole point of doing that. Authors keep talking about the improvement in the PTI measurements for 2018 deployment given improved PTI data is not included in this study. I would recommend either using the data from 2018 PTI deployment or don't repeat the statement.

**Thank you for the comment. This paper focuses exclusively on the 2016 deployment to allow for detailed case studies with the ER2-based instruments. Unfortunately, this limited us to the 2016-only PTI data. We believe, in the interest of completeness, it is valuable to examine the measured and archived 2016 PTI data within the context of the current paper, as it was a measure available during this campaign. However, we do agree with the reviewer's comment that given the QC state of the PTI data in 2016, the issues with these data should be made clearer. Following some internal discussion, we have reworded some of the text (p. 13, Line 13 – p. 14, Line 2) to more explicitly address the shortcomings with the 2016 PTI data. We have also moved the former Section 3.1/Figure 4 to the Appendix to de-emphasize its role, given its more suspect QC status relative to the other instruments. We believe it fits well in there, as it still presents a valuable comparison with the PSAP which places these data into context.**

Increase in SSA at longer wavelengths in two 80m scan is concluded due to sea salt in the boundary layer. This fact is not evident in the AAOD plot (Figure 6a). AAOD for 4STAR (80 m) scan is larger than 1 km scan indicating there were absorbing aerosol on boundary layer too.

**Thank you for this comment.  The 4STAR is looking at the entire column above-aircraft, so the difference between the two does not mean that there was additional absorbing aerosol in the boundary layer specifically.  In other words, since 4STAR sees all aerosol above the aircraft, the 80m scans are seeing boundary-layer aerosol in addition to the higher-altitude absorbing aerosol (original manuscript, p. 18, Line 30).  Based on the in situ measurements observing only scattering and negligible absorption (original manuscript, p. 18, Lines 30-31), we infer that there was no significant absorbing aerosol within the boundary layer.  The HSRL-2 data also support a pure marine aerosol classification within the boundary layer.  The differences in AODs between the two sets of 4STAR sky scans may be due to some variability (~100s of meters, as seen by HSRL-2) in the geometric thickness of the plume, and/or measurement uncertainty.  Due to this confusion, we have edited this section to make these points more clear (revised manuscript, p. 19, Lines 22-30; p. 20, Lines 33-35).**

Why RSP is different than 4STAR 80 m? As expected, SSA from in situ may not exactly be correlated with remote sensing given an impact of RH, why two remote sensing method does not correlate at all (Figure 9c)? Authors should explore that in more depth than just present the result. I am not much worried about the inconsistency between in situ vs remote sensing but why AirMSPI vs 4STAR does not show correlation with any of the optical properties discussed in this study? For example, AAE measured by 4STAR shows variations between 0.8 to 1.8 while AirMSPI AAE lies between 1.2 to 1.4. Since their conclusion for this variation is higher uncertainty in derived AAE, it is better to do in depth uncertainty analysis than just to mention. Given the fact that tow remote sensing method has more inconsistency than the remote sensing and in situ, I recommend the author to explore similar relationship (as shown in figure 10) between remote sensing methods. What is the whole point of choosing remote sensing and in situ? Given in situ have unclear bias from the RH as discuss later by the authors. In addition, the difference should be mention in relative not in absolute.

**Thank you for this comment.  There may be a number of reasons why the remote sensing instruments report different values; i.e. 4STAR differs from RSP in Figures 5 and 6, as it also differs from AirMSPI in Figures 7 and 8).   Indeed, exploring these differences is a main goal of this paper, and was previously discussed in some detail in Section 3.2.2 (original manuscript, p. 24).  Each of the SSA products presented have different advantages and limitations, and make different assumptions for their given retrievals.  In addition, the instruments will all differ in terms of instrument-specific errors, aerosol model assumption, a priori, and retrieval methods (a priori and retrieval methods play an important role when information content in the measurements is insufficient).**

 **For example, for AirMSPI, the polarized radiances measured in the 470, 660, and 865 nm bands have less sensitivity to aerosol refractive index and coarse mode aerosol size than to aerosol loading and cloud microphysical properties. As a result, the AirMSPI retrieval accuracy of refractive index and coarse mode aerosol size distribution are subjected to greater measurement errors than the AOD and cloud microphysical part of the retrieval, which potentially leads to errors with SSA (cf. the error bars with SSA).  Another contributing factor could be the effect of viewing geometry: it is not immediately evident that the information content of an observation above-aerosol looking down would be equivalent to that from an observation from below-aerosol looking up.**

Additionally, it is important to note that there are only 9 cases with both 4STAR and AirMSPI (compared with 19 for 4STAR vs in situ). Thus, due to data availability, as well as the frequent study of in situ and AERONET retrievals of SSA in the literature, these are the two upon which we focus.

Regarding AAE specifically, considering that the AAE is function of SSA and AOD, and AAOD is a smaller fraction of AOD than is the scattering AOD, the uncertainty of AAE is more impacted by the uncertainties of SSA and AOD. This leads to the observations of lowest consistency in AAE derived from 4STAR and AirMSPI in panels j-i of Figure 9.

Contributing to the above factors, a simple explanation may also be that due to the physical inability to get directly co-located aerosol conditions seen by each instrument, the different instruments are simply seeing different amounts of aerosol loading, despite our efforts to get as close a co-location as possible. This may be a particular factor when comparing 4STAR with RSP or AirMSPI, as these instruments were on different aircrafts. However, differences in scene likely do not explain e.g. the systematic differences seen between 4STAR and AirMSPI, which are likely due to differences in the retrievals themselves. Disentangling the relative effects of each of the above factors, e.g. by applying the exact same aerosol model assumption, same a priori and same retrieval method, and obtaining much tighter comparison conditions, remains to do in future by, e.g. starting from canonical case studies, and is beyond the scope of the present study.

We have revised the text (p. 38, Lines 18-32) to add more discussion of the potential causes of the observed differences.

Minor Comments:

Page 2 Line 12: Mixing state is also an important factor.

Agreed, this has been added (p. 2, Line 12).

Page 2 Line 18: Absorbing aerosol also shown to impact the global circulation patterns (Brown et al., 2018) in addition to the facts included in line 16-20.

We thank the reviewer for suggesting this interesting reference providing another example of the potential effects of absorbing aerosols on large-scale atmospheric dynamics. It has been added (p. 2, Line 20)

Page 2 Line 23: SSA is defined already.

Thank you for pointing this out. This has been revised to be more concise (p. 2, Lines 22-24).

Page 2 Line 24: "Typically between 0.7 and 0.95" this statement is not always true for BB smoke. There are lots of laboratory and ambient studies which shows a huge range of SSA from BB emissions (Liu et al., 2014; Pokhrel et al., 2016; Vakkari et al., 2014).

We thank the reviewer for these helpful references which show that much lower SSAs can be seen under conditions of high modified combustion efficiencies. We have revised this section (p. 2, Lines 26-29)

Page 11 Line 29: Is there a specific reason to calculate SSA by extinction weighted? What is the benefit of it over SSA calculated from vertically integrated absorption and scattering coefficients?

**There isn't a benefit to one method over the other, as they both give the same answer. The idea was to reproduce from the in-situ measurements what would be calculated from column-integral (i.e. remote sensing) instruments. Either calculation would accomplish this; we chose to use the extinction-weighted SSA due to the ease of this method with the available in-situ measurements.**

Page 12 Line 10: What was the noise level of 30s averages PTI?

**This information has been added (p. 13, Lines 23), in addition to the other revisions made to the discussion of the PTI data, as described in the response above to Major Comment #1.**

Page 17 Figure 4: What is the slope of the fitting line?

**This has been added to the figure caption.**

Page 18 Line 23-24: Since bars on the figure represent 10-90th percentile ranges. Which factor (instrument uncertainty or variations in data) contribute more at a longer wavelength.

**The quoted sentence refers to the uncertainty in the 4STAR sky scans, which are not given by percentile ranges in this figure but rather according to the process described in Section 2.1.1; namely, this is the range in retrieval output from a perturbation in both AOD and radiance inputs. The text has been edited to clarify this point (p. 20, line 7).**

Page 19 Line 33: 4STAR 1km plume only or sky scan?

**The 1km values referred to throughout this section are all from the sky scans performed at 1km, referred to as "plume only" to contrast with the 80m sky scans including the influence of sea salt, as described in the text. This distinction has been clarified in the text (p. 19, line 22-23).**

Page 21 Figure 5: Please use the same legend in Figure 5 & 6.

**These legends cannot be exactly the same, as they are showing different parameters (specifically for RSP and in situ instrumentation). However, the order of each of the instruments has been standardized to alphabetical in all figure legends.**

Page 23 Line 24-26: More explanation for possible reasons.

**We believe this comment is very similar to Major Comment #3 above; please see the response and revised text referenced above.**

Page 23 Line 31: How aerosol loading impact the spectral dependence of SSA?

**This sentence refers to the dependence of the in situ SSA spectral shape on the PSAP correction scheme used. Within this context, the aerosol loading may change the spectral dependence to a greater degree for the case from the 20[th] because the observed artifact (higher absorption coefficients at 470nm) would become more pronounced under higher total aerosol loading and thus total PSAP filter loading, which is the root cause of the PSAP correction artifact described in Appendix A.**

Page 23 Line 32: Do PTI and PSAP SSA's comparable at higher altitude cases only?

**No systematic differences in PSAP versus PTI data were observed for various flight altitudes. However, the PTI's instrument problems limited it to largely in-plume level legs, rather than full profiles, which could result in a sampling bias. This has been clarified in the text (p. 24, Line 35-p.25, Line 11), and the text for the Figure 4 (now Fig A3) caption has been edited to reflect this.**

Page 24 Line 3: Why AirMSPI uncertainty increase by a huge amount from about 450 to 470 nm?

**AirMSPI uncertainties are only reported for the wavelengths which are used as inputs for the retrieval runs (p. 10, Line 28). This has been clarified in the caption of this figure.**

Page 25 Figure 7: Be consistent between graph and legend.

**The items in the legend in Figure 7 are now in alphabetical order, consistent with the edits to Figures 5 and 6 requested above.**

Page 31 Line 9: Please express difference in relative.

**Thank you for this comment. We worry that the "relative" difference (we assume this refers to, e.g. taking the difference between the two as a fraction of the 4STAR SSA) is misleading here, since SSA itself is already a "relative" value (i.e., the amount of scattering relative to the total extinction). Expressing the SSA difference as a percent can be somewhat misleading, as the values are constrained between 0 and 1, and effectively further constrained between ~0.7 and 1. Consider the difference between the SSA (scattering as a fraction of total extinction) versus co-albedo (instead \*absorption\* as a fraction of total extinction; 1-SSA): an absolute difference of 0.85 vs 0.88 in SSA would be 3.5%, but in co-albedo these same measurements (now 0.15 vs 0.12, with the same 0.03 in absolute difference) give a difference of 20%. This is a limitation of SSA as a metric. Since the SSA at 530nm only range between ~0.87 and 0.81, Figure 11 (to which this comment refers) expressed in relative difference instead (as defined above) looks almost identical to the absolute difference plots. For these reasons, we think it best to leave this passage in terms of absolute SSA difference, rather than % differences, as the latter may be confusing to readers.**

**Anonymous Referee #2**

The paper presents results from several instruments to measure or retrieve aerosol optical properties, especially aerosol absorption, single scattering albedo, and their wavelength dependency during the NASA ORACLES campaign in September 2016.

The paper is well written, it is definitely worth publishing in ACP. Actually, I did not find any errors but some points I don't understand, see below. I also have some suggestions for improving the paper.

I tried to find basic info of the flights:

- Approximate flight altitude range of the two aircraft

- Thickness of plumes, flight time in plume

- Estimated age of the aerosol at the measurement location since emission from the fires

- Distance from coast and fires

- Range of scattering and absorption coefficients, RH, and AOD in the plumes

The info is within the text but very scattered. Consider giving these in a table where you present descriptive statistics of these properties. And combined to your Fig. 3.

**We thank the reviewer for this comment, but believe that the suggested table would be beyond the scope of this paper. The present work is not intended to be an overview of the ORACLES as a whole, and we worry that such a detailed summary would detract from the main message of the paper, which is the observed SSA values from each instrument overall, and the comparison between them for specific, defined case studies. A table of additional parameters should focus only on the present case studies, and runs the risk of being misinterpreted as representing the campaign as a whole, when in fact many of these conditions (e.g. plume thickness, distance from coast, estimated aerosol age) were observed to be variable beyond the specific cases presented here, and will all be the subject of additional papers as they are interesting and complex scientific questions in their own right.**

**A more detailed overview of the ORACLES campaign and overview observations will be found in a forthcoming overview paper led by the ORACLES principal investigators. Nonetheless, we have clarified the location of some of this information in the campaign description paper (Zuidema et al., 2016, https://doi.org/10.1175/BAMS-D-15-00082.1), p. 8, Lines 10-11. We also anticipate that the present ACP/AMT special issue will host many of these more specific results as they become ready for publication.**

There is the section " 4.2 Impacts of humidification on aerosols". The discussion concentrates on SSA, that discussion is ok. However, you used the in situ instruments also for calculating AOD. It is definitely underestimated by using dry in situ data. Actually, when I look at Figs. 6 and 8 it also looks like that. Discuss this. And I expected to see some figures on data collected with the RH-nephelometer. Why don't you use that? Some figure? How does the scattering enhancement vary in the plumes? Effects on AOD?

**Thank you for this comment. We hope we have adequately justified our use of the dried scattering + absorption data in our calculation of SSA; in the interest of being consistent in all of our figures, we have thus continued to use the dried scattering and absorption in Figs 6 and 8. For panels a) in these figures, the focus is on absorption only, and the effect of humidification on absorption is not well understood (Section 4.2), but we do take the point that panels b) the in situ "AOD" would be affected by humidification of the scattering. We have revised the text to quantify this difference and discuss the causes of the lower in situ "AOD" in this figure (p. 21 Lines 22-30).**

Detailed comments

P10L24, "... operational modes: step-and-stare view mode and sweep mode, " explain with a couple of sentences, not every reader knows these

**More detail has now been added to this section (p. 11, line 22-28).**

P11L22-23 "SSA was calculated using the measured PSAP absorption combined with dried (RH<40%) TSI nephelometer scattering interpolated to PSAP wavelengths." Shouldn't you use ambient RH scattering when comparing with the remote-sensing instruments? The same applies to calculating AOD from the in situ data.

**Thank you for this comment. We intended to make it clear in our paper why we have treated the humidity of the in situ instruments as we have, but we apologize that this was not adequately communicated. We have edited the text (p. 13, Lines 5-7; p. 21, Lines 22-30) to clarify this point. The SSA as well as AOD-proxy values involved are calculated using both the PSAP and the Nephelometer, and the former is by necessity a dried value. We discuss in the paper (Section 4.2) some previous work indicating that humidification may impact not just scattering, but absorption values as well. As the goal of this paper is to assess how these values compare with the other instruments, and SSA will include the effects of humidification on both components, possibly to opposing signs, we have made the decision to present the dried values for SSA to minimize this potential effect. We believe this is consistent with previous work; e.g. Haywood et al (2003) considered the impacts of RH on their in-situ-based SSA estimate to be minimal and did not correct for it.**

P12L12 "... and absorption (AAE) are calculated from PTI absorption for data within ... " How can you calculate AAE from a one-wavelength instrument? I must be missing something.

**Thank you for finding this oversight. This sentence should have read "the extinction-weighted SSA are calculated from PTI absorption…" referring to how the absorption coefficient (as shown in Figure 4) is averaged to calculate the SSA in Figure 7. The PTI does not appear in Figure 8, and AAE is not calculated for it. This has been corrected (p. 13, Lines 27-28).**

P13L19 " ... AOD, and absorption is then derived as the change in net irradiance over the AOD differential ... " Don't understand. Isn't the change of net irradiance due to both scattering and absorption? If I am wrong, please add a short explanation.

**There was some confusion in the wording; per definition, the absorbed irradiance is the difference of F_net_top - F_net_bottom. However, instead of using pairs of upwelling and downwelling irradiance at the top and the bottom of the layer as in previous studies, we instead use the full profile of the upwelling and downwelling irradiance, plotted versus the optical thickness as vertical coordinate and fit the upwelling and downwelling irradiance as follows:**

**$F^{\uparrow}$= a0 + a1*AOD(z)**

**$F^{\downarrow}$= b0 + b1*AOD(z)**

**Where a0, a1, b0, and b1 are the linear fit coefficients.**

**We then obtain the absorbed irradiance as follows:**

**F_abs = $(F^{\downarrow}_{z\_top} - F^{\uparrow}_{z\_top})$ - $(F^{\downarrow}_{z\_bot} - F^{\uparrow}_{z\_bot})$ = a0-b0 - (a0+a1*$AOD(z\_bot)$ - b0 - b1*AOD(z_bot)) = (b1-a1)*AOD(z_bot)**

**We changed the wording of the sentence in question as follows to make this more clear (p. 15, Lines 6-9): "A linear fit is performed on both upwelling and downwelling irradiance, using AOD as vertical coordinate. The absorption is then derived from the difference of the net irradiance at the top and the bottom of the layer."**

Fig 3. Name the aircraft in the Fig. Draw approximate vertical and horizontal scales in km: plume and flight altitudes, plume thickness.

**More description has been added to this figure caption. This figure is a simple schematic for the purpose of illustrating the viewing geometries used in this work, and indeed the potential viewing geometries of any similar comparison study, either over the SEA or elsewhere. Thus, we think it best to not overly define each component; for example, a similar comparison could be performed with the same instruments installed in another low-sampling aircraft aside from the P-3; for climatological conditions that show higher or lower plume altitudes; or ideally in a closer horizontal co-location than the upper bounds for separation that we give in the text. As we discussed in the overview comment from Reviewer 2 above, the plume height, cloud height, and geometric separation between the two of them was rather variable during the ORACLES deployments; thus, we think it best to not prescribe numbers on a simple illustrative schematic such as this. We have clarified the general climatological occurrence of plume heights in Section 1.1 (p. 4, Line 2).**

P17L9 "... The generally lower PTI absorption could be due to several factors..." How do you know which instrument to trust?

**Thank you for this comment. As we describe in the response to Reviewer 1, we have included additional discussion of the PTI, in particular emphasizing the quality of its 2016 data as it relates to data from other instruments (revisions on p. 13, also described above). Following this discussion, we have made the decision to remove the PTI from this comparison figure, to avoid the impression that these data are of similar quality to the other instruments presented. As discussed in the response to Reviewer 1, we retain the PTI vs PSAP comparison plot as justification for this decision. The text accompanying that figure may be found in Appendix A.2 and has been revised as well.**

In section "3.2.1 Case study: 12 September 2016" the in-situ-derived AOD is clearly lower than that derived from the remote sensing methods (Fig. 6). It is explained by the inlet cutoff on P19. Then in section "3.2.2 Case study: 20 September 2016" the AODs are essentially the same (Fig. 8). Why is that? Why couldn't the explanation for 9/12 be an aerosol layer above the plume measured with the in situ instruments? Have you considered comparing differential AODs of the in situ and remote-sensing methods? That would help in eliminating possible error due to high-level aerosol not observed with the in situ instruments.

**Thank you for this comment. Our response is similar to that for Reviewer 1, Major Comment 2. As we attempted to convey within the manuscript, we have several lines of evidence pointing towards the contribution of boundary-layer marine aerosol, and the absence of extra high-level aerosol, in the case on 12 September (and on the 20[th], as well). In addition to the observation of large particles within the boundary layer (as mentioned by the reviewer), we make use of the HSRL-2 lidar data, on the ER-2 aircraft. The HSRL-2 shows the plume height during the overpass as 2.25-5.5km (original manuscript, p. 12, Line 23) and saw no higher aerosol layer than that plume during this case. Given that the profile of the in situ extends to 5.8km, and the in situ observations, we are confident that the majority of the plume is included in this analysis.**

**We also note that the case on the 20[th] exhibits a somewhat greater range of AODs from the different instruments, likely due to a change in the plume top height, again as observed by the HSRL-2, though still below the maximum profile altitude of 7.1km in this case.**

**We have clarified the altitude limits of the in situ profiles shown, and the HSRL-2 observations which show no higher aerosol in either case (p. 19, Line 25 and p. 24, Lines 16-17)**

P27L28 "..due to the small range in AAE values retrieved from AirMSPI" Any explanations or hypotheses?

**It is difficult to say definitively what caused this small range in the AirMSPI data. We note that since the AAE is function of SSA and AOD, and AAOD is a smaller fraction of AOD than is the scattering AOD, the uncertainty of AAE will be impacted by the compound uncertainties of SSA and AOD. This leads to the observations of lowest consistency in AAE derived from 4STAR and AirMSPI in panels j-i of Figure 9. This is related to the information content question discussed above; for AirMSPI, the polarized radiances measured in the 470, 660, and 865 nm bands have less sensitivity to aerosol refractive index and coarse mode aerosol size than to aerosol loading and cloud microphysical properties. As a result, the AirMSPI retrieval accuracy of refractive index and coarse mode aerosol size distribution are subjected to greater measurement errors than the AOD and cloud microphysical part of the retrieval, which potentially leads to errors with SSA which may then be compounded in the subsequent calculation of AAE. We have added some more discussion on this question (p. 29, Lines 11-15 and p. 38, Lines 18-22)**

P28L4-5 " ... values of AAE for a mixed aerosol (smoke) that are less than 1 (for the wavelength range 440-675nm) are suspect in that they run counter to theory ... " This is not true, it is easy to show with a Mie code that AAE can be < 1 both for uncoated and coated BC particles. Have a look at Figs. 7-9 of Gyawali et al. (2009) and Fig 8 of Lack and Cappa (2010) and the related discussions in these papers.

**Thank you for this comment. We had endeavored to capture these nuances in our original draft (p. 28, "While pure BC is typically considered to have an AAE of 1, studies have shown this may vary based on particle size (e.g., Lack and Cappa, 2010; Wang et al., 2016), and may be either higher or lower than this value.") but have revised to make this more clear.**

**While we still advocate caution on the part of the reported AAE values if only for being uncorrelated to one another (and on the part of 4STAR, lower than the other measurements even in the presence of brown carbon, which generally has significantly higher AAE), we have nonetheless revised the text of this passage to better reflect the expected AAE values (p. 29, Lines 11-15).**

[revised manuscript text omitted]